# From Words to Numbers: Your Large Language Model Is Secretly A Capable Regressor When Given In-Context Examples

**Robert Vacareanu**[*]
University of Arizona
Technical University of Cluj-Napoca

**Vlad-Andrei Negru**
Technical University of Cluj-Napoca

**Vasile Suciu**
Technical University of Cluj-Napoca

**Mihai Surdeanu**
University of Arizona

## Abstract

We analyze how well pre-trained large language models (e.g., Llama2, GPT-4, Claude 3, etc) can do linear and non-linear regression when given in-context examples, without any additional training or gradient updates. Our findings reveal that several large language models (e.g., GPT-4, Claude 3) are able to perform regression tasks with a performance rivaling (or even outperforming) that of traditional supervised methods such as Random Forest, Bagging, or Gradient Boosting. For example, on the challenging Friedman #2 regression dataset, Claude 3 outperforms many supervised methods such as AdaBoost, SVM, Random Forest, KNN, or Gradient Boosting. We then investigate how well the performance of large language models scales with the number of in-context exemplars. We borrow from the notion of regret from online learning and empirically show that LLMs are capable of obtaining a sub-linear regret.[1]

## 1 Introduction

Large Language Models (LLMs) are capable of learning to perform a task when given examples of that task in their context, without any additional training. This surprising capability, called in-context learning (ICL) Brown et al. (2020), emerges just from next-token prediction for sufficiently large models.

We use regression tasks to analyze the in-context capabilities of already pre-trained large language models (LLMs), such as Llama2, GPT-4, or Claude 3. Garg et al. (2022) have previously explored the range of functions that transformers, when trained specifically for in-context learning, are capable of learning. However, contemporary LLMs emerge as capable in-context learners without being specifically trained for it. We extend previous work and analyze the extent to which LLMs, decoder-only transformers trained auto-regressively for next-token prediction, are capable of learning regression functions when given in-context exemplars, **without any additional form of supervision or training**.

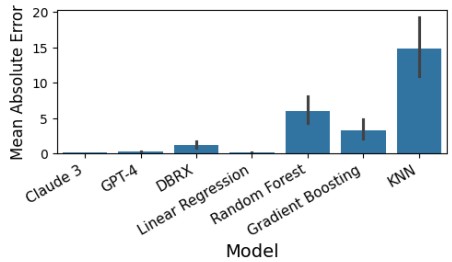

Figure 1: Mean Absolute Error ($\downarrow$) comparison between three large language models (LLMs) and four traditional supervised methods for learning a linear regression function with one informative variable out of two. Given only in-context examples and without any additional training or gradient updates, pre-trained LLMs such as Claude 3, GPT-4, or DBRX can outperform supervised methods such as Random Forest or Gradient Boosting.

---

[*]Correspondence to rvacareanu@arizona.edu

[1]Code available at https://github.com/robertvacareanu/llm4regression

Similar to previous work (Garg et al., 2022), we use (synthetic) regression datasets. Synthetic regression datasets have the following advantages:

**(i) Algorithmically generated**: The data is guaranteed to be generated deterministically, by a well-determined (and logical) formula. This property makes them suitable to use when investigating whether a given model is capable of unraveling the underlying structure of the data.

**(ii) Difficulty control**: The user has direct access to the difficulty of the synthetic regression problem and can investigate the cases of simple linear regressions of the form $y = ax + b$, to more difficult problems such as Friedman #2, a highly non-linear function used for benchmarking:

$$y = \sqrt{(x_1^2 + (x_2 \cdot x_3 - \frac{1}{x_2 \cdot x_4})^2)}$$

**(iii) Data availability**: Lastly, synthetic datasets present the advantage of allowing the user to generate novel data in large(r) quantities. Additionally, it ensures that models are less likely to have been previously exposed to these specific problems.

Formally, let $\mathbb{D}_n$ be a dataset consisting of $n$ input-output examples: $\mathbb{D}_n = \{(x_1, y_1), \ldots, (x_n, y_n)\}$, where $x_i \in \mathbb{R}^d$, with $1 \le d \le 20$ typically. We have $y_i = f(x_i)$, where $f : \mathbb{R}^d \to \mathbb{R}$ and $y_i \in \mathbb{R}$. We do not put any restrictions on $f$ and study functions ranging from simple linear predictions (i.e., $f(x) = ax + b$) to more complex and highly non-linear functions (e.g., $f(x) = x + 10sin(\frac{5\pi x}{100}) + 10cos(\frac{6\pi x}{100})$). We study how well various models such as LLMs (e.g., GPT-4), traditional supervised models (e.g., Random Forest), and unsupervised baselines (e.g., random prediction) are capable of predicting $y_{n+1}$ when given access to $n$ input-output examples (i.e., $\mathbb{D}_n$) and $x_{n+1}$.

tOur study shows that pre-trained large language models (LLMs) display a surprisingly good performance on various regression tasks. For example, in Figure 1, **without any parameter update**, Claude 3 approaches the performance of a Linear Regression model and largely outperforms other supervised methods such as Random Forest or Gradient Boosting on a randomly generated linear regression dataset, with one informative variable out of two.

## 2 Experimental Setup

We describe the models and the datasets we use in our experiments below.

### 2.1 Datasets

We experiment with 3 types of datasets: (1) linear regression, (2) non-linear regression, and (3) regression datasets with non-numerical inputs. We describe each below.

#### 2.1.1 Linear Regression Datasets

We experiment with linear regression tasks of the form $y = wx + b$, where $w, x \in \mathbb{R}^d$, $b \in \mathbb{R}$, and $y \in \mathbb{R}$, with $1 \le d \le 10$. We vary both $d$, the dimension of the input $x$, and the number of *informative* variables (i.e., the number of non-zero elements in $w$).

When generating a dataset, we sample the input $x$ from $\mathcal{N}(0, 1)$. We sample the weight vector $w$ from $Uniform(0, 100)$.[2]

#### 2.1.2 Non-Linear Regression Datasets

For non-linear regression problems, we use the three problems introduced by Friedman, called `Friedman #1`, `Friedman #2`, and `Friedman #3` (Friedman, 1991; Breiman, 1996). For

---

[2]We used sklearn. Please see `make_regression` for more details.

example, `Friedman #1` is defined as follows:

$$y(x) = 10 * sin(x_0 x_1 \pi) + 20(x_2 - 0.5)^2 + 10x_3 + 5x_4 + \epsilon * N(0,1)$$

Where $y \in \mathbb{R}, x \sim Uniform(0,1)^d$ and $\epsilon \in \mathbb{R}$. We have $5 \leq d$. When $d > 5$, the extra dimensions are ignored for the computation of $y$.

While we create these datasets with different random seeds, resulting in different $\mathbb{D}_n$, making a particular $\mathbb{D}_n$ very unlikely to have been seen by the LLMs during training, it is *still* possible that they have seen different $\mathbb{D}_n$ originating from the same generator function $f$. In an attempt to mitigate this risk, we created 5 new non-linear datasets. We describe them in the Appendix C. For example, one of these functions is: $y(x) = x + 10sin(\frac{5\pi x}{100}) + 10cos(\frac{6\pi x}{100})$, where $x \sim Uniform(0,100)$ (plotted in Figure 5).

To supplement the non-linear regression datasets and following Garg et al. (2022), we create datasets using randomly initialized neural networks. We explore the outputs of 2 types of neural networks: (1) a sequence of simple linear layers with ReLU non-linearity in-between, and (2) the output of a randomly initialized transformer encoder block.

### 2.1.3 Regression With Non-Numerical Inputs

To further investigate whether the models are able to learn abstract tasks beyond those subsumed by numerical regression (Razeghi et al., 2022), we design the following tasks. We (randomly) map symbols (i.e., characters) to numbers (e.g., $a \rightarrow 1$). We then randomly sample a subset of these symbols in order to keep the context size manageable and to not need a large number of examples. We map the symbols to a numerical value by sampling a weight vector $w \in \mathbb{R}^d$ and doing a dot product between it and the corresponding values of each symbol. We use lowercase ASCII letters as our symbols (i.e., a ... z). We randomly sample 5 symbols which will serve as our vocabulary. We include the pseudocode in Appendix C.11.

### 2.2 Models

We experiment with three types of models: **(1) large language models** such as GPT-4, **(2) supervised models** such as Random Forest, and **(3) heuristic-based unsupervised models** such as random sampling. All models have access to the same train data and are evaluated on the same test partition. They have access to an input dataset $\mathbb{D}_n$ and are asked to predict the $y_{n+1}$ corresponding to the $x_{n+1}$. The train partition is used for in-context exemplars for LLMs and supervised training for supervised methods. Due to budget constraints and the context size limitations of the LLMs, we round input values to two decimal places.[3] We repeat each experiment with different random seeds.

**LLMs:** We use a total of 12 large language models (LLMs), both open and private. Specifically, we use Mistral7B, Mixtral8x7B, CodeLlama70B, Llama2 70B, Yi 34B, DBRX (*weights available*) and ChatGPT, GPT-4 (OpenAI) and Claude 3 Opus, Claude 3 Sonnet (Anthropic), Gemini Pro (Google), and Mistral Medium (Mistral) (*weights not available*). The models we use cover a wide range of parameters, from 7B or less (Mistral) to 132B or more (DBRX).[4] Unless otherwise specified, we interact with the models only through prompting and in-context exemplars. We use the same prompt for all models and do not do any prompt tuning. The prompt is of the form `Feature 1: <number>\nFeature 2: <number>\nOutput: <number>`. In-context exemplars are separated with two new lines "\n\n". For the test example, the model is asked to predict the number corresponding to the `Output` variable. We observed that some models tend to provide additional explanations, before outputting the final number. To prevent this behavior, we add an additional text in the beginning, instructing the LLM to only output the number. We give a complete example in Appendix D.1.1. Additionally,

---

[3]We provide extra experiments without rounding in Appendix N to show that the strong results we observed are not an artifact of rounding.

[4]Since the number of parameters for some models is not disclosed, it is possible that certain closed models may have fewer than 7B or more than 132B parameters.

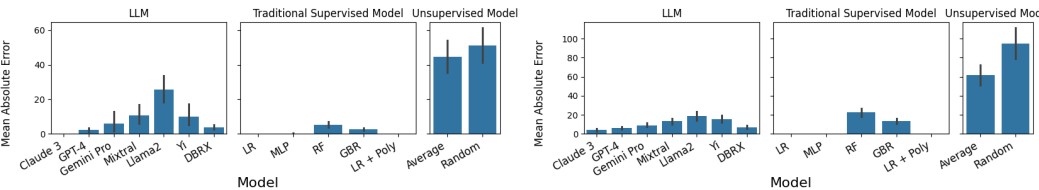

(a) Informative Variables: 1; Total Variables: 3  (b) Informative Variables: 2; Total Variables: 2

Figure 2: The performance, as measured by the Mean Absolute Error ($\downarrow$), across large language models (LLM), traditional supervised models and unsupervised models on two different random regression tasks: (a) sparse linear regression, where only 1 out of a total of 3 variables is informative, and (b) linear regression with two informative variables. The results are averages with 95% confidence intervals from 100 runs with varied random seeds. All LLMs perform better than the unsupervised models, suggesting a more sophisticated underlying mechanism at play in ICL. Furthermore, some LLMs (e.g., Claude 3) even outperform traditional supervised methods such as Random Forest or Gradient Boosting.

we analyze the explanations provided by the models in Appendix L, finding that there is sometimes a discrepancy between the rationale given for their predictions and the actual predicted values. Unless otherwise specified, we use a temperature of 0.

**Supervised Baselines:**    We use a total of 10 traditional supervised models, available in most statistical learning packages. We use: Linear Regression (4 versions: (i) no regularization, (ii) Ridge regularization, (iii) Lasso Regularization, and (iv) no regularization and with polynomial features), Multi-Layer Perceptron (6 versions, 3 versions with different widths (Cybenko, 1989; Hornik et al., 1989) and 3 versions with different depths (Lu et al., 2017)), Random Forest, Bagging, Gradient Boosting, AdaBoost, SVM, KNN, Kernel Ridge, and Splines. Similar to the LLM case, we do not tune any hyperparameters and use the defaults available in sklearn. It is important to note that these supervised baselines are very strong: (1) many of them are the results of algorithms specifically designed for regression (e.g., Splines); (2) all perform parameter updates (unlike an LLM with ICL); and (3) the default hyperparameters, as set in widely-used statistical packages, have been refined over time to offer reliable and generally strong performance across a variety of scenarios.

**Unsupervised Baselines:**   In order to contextualize the performance of the LLMs and to evaluate their effectiveness relative to basic heuristics, we incorporated the following series of heuristic-based unsupervised baseline:

1. **Average:** Predicts the next value, $y_{n+1}$, as the mean of all preceding outcomes: $y_{n+1} = \frac{1}{n} \sum_{i=1}^{n} y_i$.
2. **Last:** Uses the most recent tuple $(x_n, y_n)$ for prediction, such that $y_{n+1} = y_n$.
3. **Random:** Predicts $y_{n+1}$ by randomly selecting from the set of prior observations $\{y_1, \ldots, y_n\}$. The final prediction is thus $y_{n+1} = sample([y_1, \ldots, y_n])$

Additional details on the models are provided in Appendix D. We include results with additional models, such as the latest release of GPT-4 at the time of submission (gpt-4-2024-04-09) or Mixtral Mixture of Experts 8x22B in Appendix G, where we present the average rank obtained by each model.

## 3   Large Language Models Can Do Linear Regression

Our first experiment intends to capture how well LLMs can do linear regression when given only in-context examples. To this end, we experiment with 4 different types of regression problems, varying the number of total variables and the number of informative variables. We provide a total of 50 tuples $\mathbb{D}_{50} = \{(x, y)_i \,|\, i = 1 \ldots 50\}$ as in-context exemplars and

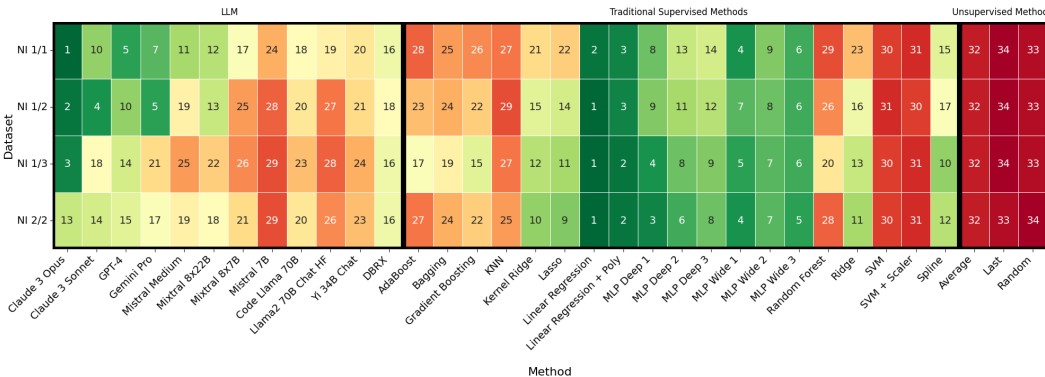

Figure 3: The rank of each method investigated over all four linear regression datasets. Rankings are visually encoded with a color gradient, where green means better performance (higher ranks) and red indicates worse performance (lower ranks). Notably, very strong LLMs such as Claude 3 and GPT-4 consistently outperform traditional supervised methods such as Gradient Boosting, Random Forest, or KNN. (best viewed in color)

ask the model to generate $y_{51}$, corresponding to $x_{51}$. We repeat the experiments with 100 different random seeds and present the average and 95% confidence intervals.

We present two bar plots in Figure 2, corresponding to two different datasets: **(1)** a dataset consisting of three variables, with a single informative variable (`Regression NI 1/3`), and **(2)** one dataset containing two random variables, where both variables are informative (`Regression NI 2/2`). For LLMs, we selected Claude 3 Opus (Claude 3), GPT-4, and Gemini Pro, as they are the flagship closed-source models currently available, and Mixtral8x7B (Mixtral), Llama2 70B (Llama 2), Yi 34B (Yi) and DBRX (DBRX) as the flagship open-weights models. Traditional supervised models in our analysis included Linear Regression (LR), Multi-Layer Perceptron (MLP), Random Forests (RF), and Gradient Boosting (GB). Additionally, we include a fifth supervised method, the one resulting in the best performance.[5] We would like to remark that this is a **very** strong baseline, as it highlights the best performance obtainable with hindsight information. For the unsupervised baselines we included (i) Average, and (ii) Random Sampling. We draw the following observations from this experiment:

First, LLMs, when given in-context examples of input-output pairs, exhibit a (perhaps surprisingly) good overall performance. When compared with unsupervised baselines, the large language models **always** outperform them, indicating that the underlying mechanism at play is more sophisticated than such simple heuristics.

Second, we remark that LLMs in some cases outperform even supervised methods. For example, for the regression task with one informative variable out of a total of 3 (`Regression NI 1/3`), Claude 3 ranks 3 out of a total number of 31 models, only (slightly) behind `Linear Regression` and `Linear Regression + Poly`. For example, when averaging the mean absolute error across all runs, Claude 3 obtains 0.14, while `Linear Regression` obtains 0.12. It largely outperforms other supervised methods such as `Random Forest` or `Gradient Boosting`, even though it no gradient updates were performed, nor it was specifically designed for linear regression.[6]

Lastly, we remark that this strong performance is not only specific to the current closed-source flagship models. For example, Mixtral outperforms supervised methods such as Random Forest or Gradient Boosting on the `Regression NI 2/2` dataset.

Alongside the two bar plots, we include a heatmap in Figure 3 to show how each model ranks across different datasets. We show the datasets vertically and the models horizontally.

---

[5] If this method coincides with one previously selected, the subsequent best performer is chosen.

[6] Comparatively, `Random Forest` obtains 5.32, `Gradient Boosting` obtains 2.58, and GPT-4 obtains 2.26.

For instance, Claude 3 Opus achieves the top rank (rank=1) on the NI 1/1 dataset. Notably, Claude 3 Opus and GPT-4 consistently perform better than methods such as AdaBoost, Gradient Boosting, KNN, or Random Forest. Out of the LLMs with open-weights, both Mixtral 8x7B and Yi 34B Chat outperform methods such as KNN or SVM on all four datasets.

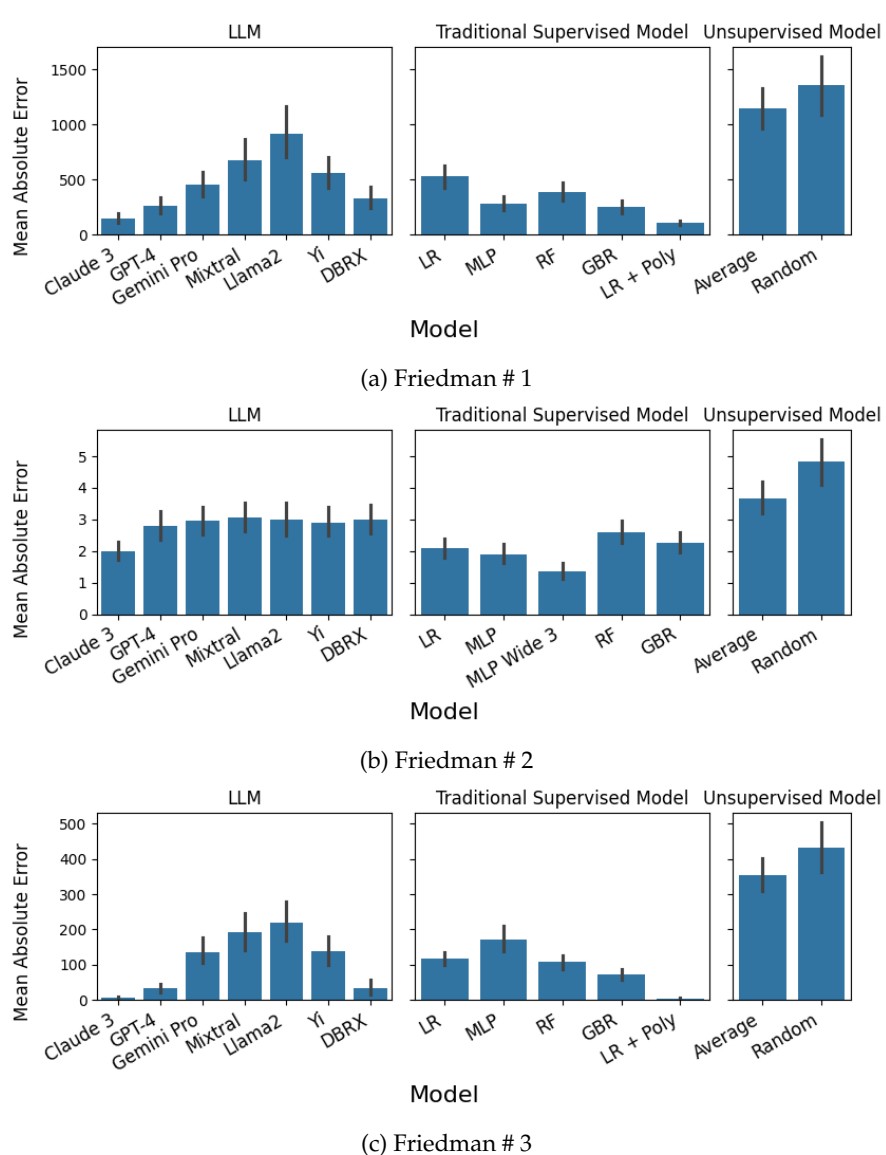

(a) Friedman # 1

(b) Friedman # 2

(c) Friedman # 3

Figure 4: The performance of large language models (LLM), traditional supervised models and unsupervised models on Friedman #1, #2, and #3. The results represent the averages with 95% confidence intervals over 100 different runs.

Overall, these results reveal that large language models, whether closed-source (e.g., Claude 3, GPT-4) or open-weights (e.g., DBRX, Mixtral 8x7B), are capable of performing linear regression tasks using in-context exemplars composed of $(x, y)$ pairs, all without the necessity for gradient updates. While the performance across these models varies, it consistently outperforms that of unsupervised baselines, suggesting that the underlying mechanism at play is more sophisticated than these simple heuristics. Moreover, specific LLMs (e.g., Claude 3, GPT-4) consistently exceed the performance of strong supervised baselines such as Random Forests, Gradient Boosting, or KNN.

We present extended results, encompassing a wider array of models and datasets, in Appendix E.

## 4 Large Language Models Can Do Non-Linear Regression

We extend our previous analysis to non-linear regression problems.

### 4.1 Friedman Benchmarks

We use the 3 synthetic regression benchmarks introduced by Friedman (1991). Below, we provide the definition of the Friedman #2 dataset, with complete definitions for all datasets available in Appendix C.

$$f(x) = \sqrt{x_1^2 + \left(x_2 \cdot x_3 - \frac{1}{x_2 \cdot x_4}\right)^2} + \epsilon \cdot \mathcal{N}(0, 1) \tag{1}$$

where $\epsilon$ represents noise added to the system, modeled as a Gaussian distribution $\mathcal{N}(0, 1)$, and the variables $x_1$, $x_2$, $x_3$, and $x_4$ are drawn from uniform distributions as follows: $x_1 \sim \mathcal{U}(0, 100), x_2 \sim \mathcal{U}(40\pi, 560\pi), x_3 \sim \mathcal{U}(0, 1)$, and $x_4 \sim \mathcal{U}(1, 11)$.

Our findings for the Friedman #1, #2, and #3 benchmarks are presented in Figure 4. The selection of methods follows to the same procedure used in Section 3: three leading closed-source LLMs, four leading open-weights LLMs, and five conventional supervised models–including the best performing model–and two unsupervised baselines. We remark that the strong performance of LLMs persists in the non-linear case as well. For example, Claude 3 outperforms all but the Linear Regression with Polynomial Features (`LR + Poly`) on Friedman #2.

### 4.2 New Regression Datasets

In an effort to mitigate the potential familiarity of models with pre-existing datasets encountered during their pre-training phase, we experiment with two new non-linear regression datasets which are unlikely to have been part of the pre-training phase. Our methodology is as follows. Our first novel dataset (called `Original #1`), plotted in Figure 5, is created to resemble a line with oscillations:

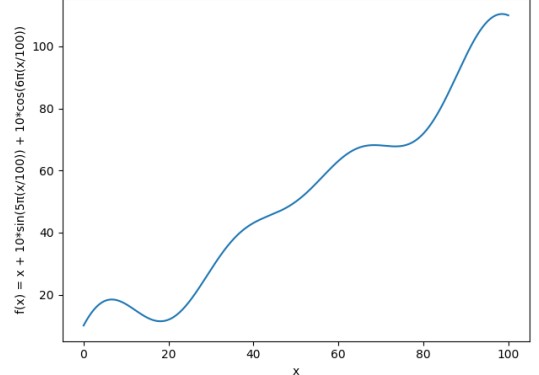

$$y = x + 10sin(\frac{5\pi x}{100}) + 10cos(\frac{6\pi x}{100}) \tag{2}$$

Where $x \sim \mathcal{U}(0, 100)$.

Figure 5: An example of one of our new non-linear regression functions. The function was designed to mimic a linear trend with oscillations.

For the next dataset (called `Original #2`), we draw inspiration from the datasets introduced by Friedman, but we modify the domain of $x$ and change the operands (e.g., $^2 \rightarrow {}^4$). We provide an example below:

$$y = (x_1^4 + (x_2 \cdot x_3 - \frac{2}{\sqrt{x_2} \cdot \sqrt{x_4}})^2)^{\frac{3}{4}} \tag{3}$$

It is important to underscore that the primary goal of these novel datasets is not to construct inherently difficult challenges for the LLMs, but rather to minimize the probability of

evaluating them on datasets they could have already seen during their training phase. We provide additional details on these datasets in Appendix C, along with additional datasets. For an in-depth analysis of potential data contamination concerns, including additional experiments conducted to address these issues, please refer to Appendix P.

### 4.3 Discussion

We summarize all our results in the form of a heatmap in Figure 6. For each dataset, we record the relative rank of each method with respect to all the others. For example, Claude 3 Opus performs the best on Original 1 (rank=1). We structure our results in 3 blocks: (1) LLMs (left), (2) Traditional Supervised Methods (middle), and (3) Unsupervised Methods (right). We make the following observations:

First, on Original 1 (see Figure 5), LLMs largely outperform traditional supervised methods. Remarkably, eight out of the ten highest-ranking methods in this context are LLMs. This strong performance on this dataset is exhibited by both private and open-weights models. For example, DBRX outperforms all traditional supervised methods, despite no gradient update.

Second, we remark that the LLMs show a strong performance on all datasets introduced by Friedman (Friedman #1, Friedman #2, Friedman #3) and on all datasets introduced by us (Original #1, Original #2).

Overall, our results show that **LLMs with ICL are capable of performing non-linear regression**. For example, Claude 3 Opus outperforms Gradient Boosting and KNN on all 5 datasets. We present extended results, encompassing a wider array of models and datasets, in Appendix F. We observed that LLMs struggle on the datasets generated with randomly initialized neural networks (e.g., Simple NN #1, Transformer #1), although they remain, generally, better than the unsupervised methods. Due to space constraints, we included in Appendix M an analysis of the performance of LLMs on non-numerical regression datasets. We found that even in this regime, LLMs outperform the unsupervised baselines.

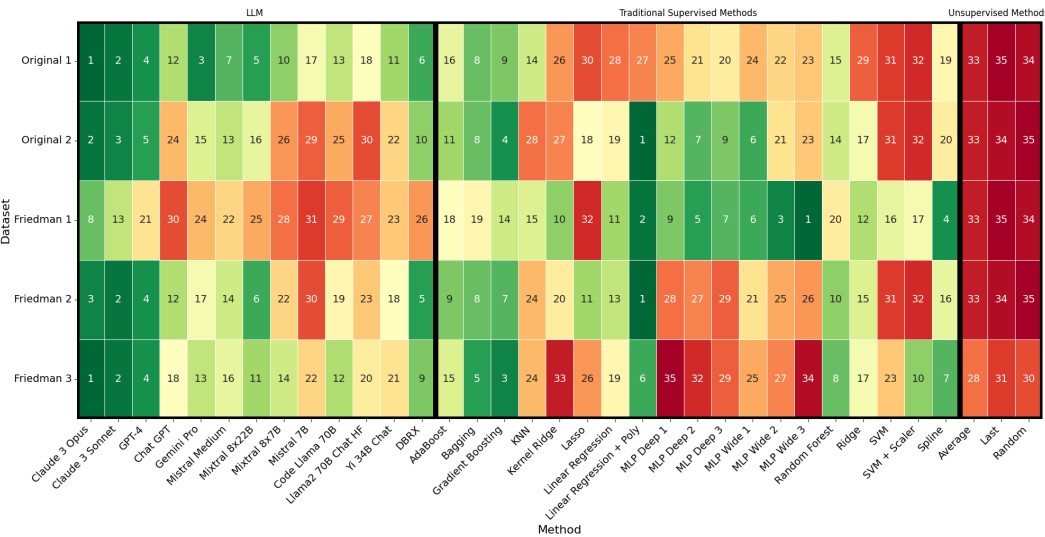

Figure 6: Rank of each model investigated on the non-linear regression datasets. LLMs are capable of non-linear regression. For example, for Original #1, eight out of the ten highest-ranking methods are LLMs. (best viewed in color)

## 5 How Fast Do Large Language Models Adapt?

Following the surprising results that LLMs are capable of doing regression, when given the training data in their context in the form of in-context exemplars, we investigate next how

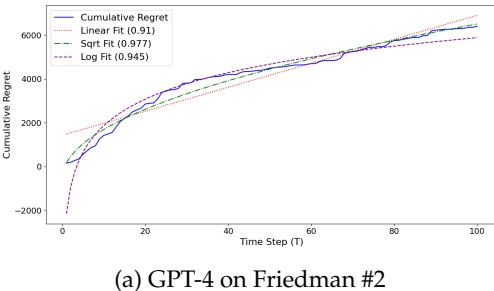
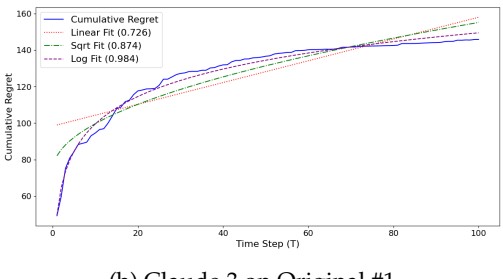

| (a) GPT-4 on Friedman #2 | (b) Claude 3 on Original #1 |

Figure 7: The cumulative regret of two large language models on two different non-linear regression dataset. Both show a sub-linear regret grow, indicating that as more data points are observed, the models become increasingly efficient at predicting outcomes closer to the optimal strategy derived in hindsight.

their predictions improve with the number of examples given. Specifically, we empirically analyze whether the performance of the LLMs approaches that of the best possible fixed strategy over time.

Borrowing from the Online Learning community, we empirically analyze how the cumulative regret (i.e., cumulative loss) grows with respect to the time step (number of examples in context) (Orabona, 2019). Ideally, a good model should, over time, approach the quality of decisions that the best fixed strategy, informed by hindsight, would have made. In other words, the cumulative regret should, ideally, grow sub-linearly over time. To empirically estimate how the regret grows, we fit 3 curves: (1) Linear Fit: $a * x + b$, (2) Sqrt Fit: $a * sqrt(x) + b$ and (3) Log fit: $a * log(x) + b$.[7] We then use the $R^2$ coefficient to determine which curve fit is better. We show two qualitative plots in Figure 7. We summarize the results in Table 1, recording the curve fit with the highest $R^2$ coefficient for each model. Since simply picking the best curve fit according to the $R^2$ score might tell an incomplete story, we include additional plots in Appendix H, covering multiple models and all seven datasets. We draw the following observations. First, the performance of large language models improves with the number of examples, suggesting the mechanism at play is capable of **effectively** leveraging more data. Second, we remark that very capable LLMs, such as Claude 3 or GPT-4 can obtain sub-linear regret, meaning that the predictions made by the LLM approach the quality of decisions that the best algorithm would have made, leading to near-optimal performance in the long run.

We remark that there are differences between our empirical analysis and online learning. Firstly, while online learning often focuses on establishing theoretical regret bounds, our approach is empirical, we only empirically show that the regret of certain LLMs grow sub-linearly by using curve fitting and $R^2$. To address potential concerns of overfitting and enhance the robustness of our empirical findings, we repeated the experiment 3 times and averaged the cumulative regret. Second, our results are only for finite (and relatively small) time steps, diverging from the online learning norm of analyzing behavior as $T$ approaches infinity. To provide further evidence that the results are not an artifact of small T, we performed the following experiment. We used GPT-4 and recorded its performance across multiple training dataset sizes, ranging from 20 to 500. We have observed that the performance of GPT-4 continues to improve as the number of in-context exemplars increases, suggesting that, our results are not an artifact of limited time steps. We include the associated plots in Appendix Q.

Following the empirical evidence that LLMs are very capable regressors, despite not being trained for it, we hypothesize that (very capable) LLMs emerge from their training as very good online meta-learners (Finn et al., 2019; Mirchandani et al., 2023).

---

[7]The choice of linear, square root, and logarithmic fits is motivated by their common appearance in theoretical regret bounds within the online learning literature.

| Model \Dataset | Friedman 1 | Friedman 2 | Friedman 3 | Original 1 | Original 2 | Regression NI 1/3 | Regression NI 2/2 |
|---|---|---|---|---|---|---|---|
| Claude 3 Opus | linear | sqrt | sqrt | log | sqrt | log | log |
| GPT-4 | linear | sqrt | sqrt | log | sqrt | log | sqrt |
| Gemini Pro | linear | sqrt | linear | log | sqrt | sqrt | sqrt |
| Yi 34B Chat | linear | sqrt | linear | sqrt | sqrt | sqrt | sqrt |
| Mixtral 8x7B | linear | linear | linear | sqrt | linear | linear | sqrt |
| Mistral 7B | linear | linear | linear | sqrt | linear | linear | linear |
| DBRX | linear | log | linear | log | sqrt | sqrt | sqrt |
| AdaBoost | linear | sqrt | linear | sqrt | sqrt | sqrt | sqrt |
| Gradient Boosting | sqrt | sqrt | linear | log | sqrt | log | sqrt |
| Linear Regression | linear | linear | linear | linear | linear | log | log |
| Linear Regression + Poly | sqrt | log | log | linear | log | log | log |
| Random Forest | linear | sqrt | linear | sqrt | sqrt | sqrt | linear |
| KNN | linear | linear | linear | log | linear | sqrt | sqrt |

Table 1: We show which curve-fit obtained the highest $R^2$ for multiple models and datasets. The slower the growth of the function, the better (i.e., $log > sqrt > linear$). (best viewed in color)

# 6 Related Work

Many previous papers studied in-context learning (Min et al., 2022; Dai et al., 2023; Pan et al., 2023; Kossen et al., 2024), either from a more theoretical standpoint (von Oswald et al., 2022; Xie et al., 2022; Wies et al., 2023; Ahn et al., 2023; Vladymyrov et al., 2024) or from a more empirical one (Garg et al., 2022; Akyürek et al., 2023; Zhang et al., 2023). For a more general discussion of in-context learning, we refer the reader to Dong et al. (2023). Our work fits into the empirical camp, showing through experiments that large language models pre-trained for next-token prediction on web-scale datasets are capable of doing regression when shown input-output examples in their context, sometimes with performance rivalling that of supervised methods such as Gradient Boosting or Random Forests. Previous studies Garg et al. (2022); Bai et al. (2023); Guo et al. (2024); Bhattamishra et al. (2024) have explicitly demonstrated the capacity of transformers to be trained for in-context learning, enabling them to implement a wide range of algorithms when provided with relevant examples. For example, it has been shown that transformers can be trained to in-context learn linear functions, with performance similar to that of optimal least squares (Garg et al., 2022; Zhang et al., 2023). Bai et al. (2023) extended this, showing that transformers can implement a broad class of standard machine learning algorithms in context when pre-trained for it. Li et al. (2023) proved generalization bounds for ICL under specific conditions. Different from this line of work, we do not train any model specifically for in-context learning or regression. Instead, we simply use large, powerful models (e.g., Claude 3, GPT-3, Llama2) and empirically show that they are capable of performing linear and non-linear regression when given in-context examples, despite not being specifically trained for it.

# 7 Conclusion

In this paper, we examined the extent to which large language models such as Claude 3, GPT-4, or DBRX are capable of performing the task of regression, when given input-output pairs as in-context examples, without any gradient updates.

We showed that large language models are capable of doing both linear and non-linear regression, with performance rivaling that of supervised methods such as Linear Regression or Gradient Boosting. We then analyzed how their performance approaches that of the best possible fixed strategy as the number of in-context examples grows, showing how very capable models such as Claude 3 Opus or GPT-4 are capable of approaching the quality of decisions that the best algorithm in hindsight would have made. Our results demonstrate that large language models are capable of doing regression when given in-context examples of (input, output) pairs, despite not being explicitly trained to do so. We leave the exploration of augmenting LLMs' training with synthetic regression and math datasets, during either pre-training or fine-tuning, to future work. We release our code and results at https://github.com/robertvacareanu/llm4regression.

## Acknowledgments

This work was partially supported by the Defense Advanced Research Projects Agency (DARPA) under the ASKEM and Habitus programs. Mihai Surdeanu declares a financial interest in `lum.ai`. This interest has been properly disclosed to the University of Arizona Institutional Review Committee and is managed in accordance with its conflict of interest policies.

## 8  Ethics Statement

In this work we explored the extent to which large language models (LLMs) are able to perform regression tasks. We did not perform any additional training. We do not envision any negative impact of our results.

## 9  Limitations

This study focuses primarily on regression tasks, including an exploration into regression-like scenarios where inputs are symbolic rather than numeric, yet the outputs remain numeric.

A second limitation is the reliance on several large language models, including proprietary ones whose performance may change over time, potentially affecting reproducibility. To address this, we also included leading open-weight models in our analysis, though we note that their performance is generally behind that of private models. Additionally, we release our intermediate results.

Third, the issue of data contamination poses a challenge, given the opaque nature of training datasets for many LLMs. We have taken several steps to mitigate this risk: (i) Our analysis spans multiple LLMs, reducing the likelihood of all models being contaminated in the same way; (ii) We evaluated models with multiple random seeds on newly introduced datasets (alongside known ones like `Friedman #1`). In this way, we diminish the chance that these models have been directly exposed to the exact datasets during training; (iii) We included results with Falcon 40B, whose training data is publicly available (please see Appendix P for more details). We acknowledge that these measures do not eliminate the potential for data contamination entirely.

Fourth, while we showed empirical evidence that large language models are capable to perform regression tasks, we did not provide theoretical explanations to support these observations.

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

## A Appendix Structure

We organized this appendix as follows.

In Appendix B we expand on the related work discussion.

In Appendix C we provide additional details of the datasets we used, including their underlying formulas.

In Appendix D we provide additional details of the models we used, together with their exact API code.

In Appendix E, we provide additional experiments and results to complement Section 3: Large Language Models Can Do Linear Regression.

In Appendix F, we provide additional experiments and results to complement Section 4: Large Language Models Can Do Non-Linear Regression.

In Appendix G, we show the average ranks obtain by each model across different dataset types.

In Appendix H, we provide additional experiments and results to complement Section 5: How Fast Do Large Language Models Adapt?

In Appendix I, we provide additional experiments and results on real-world datasets.

In Appendix J, we provide results with more models from the Claude family.

In Appendix K we detail the total costs of running all the large language models.

In Appendix L we detail how LLMs provided justifications for their prediction.

In Appendix M we include another experiment: regression task with non-numerical inputs.

In Appendix N we analyze the effects of rounding.

In Appendix O we analyze whether the performance of LLMs is similar with KNNs or not.

In Appendix P we analyze whether the results we have seen could be the effect of data contamination.

In Appendix Q we analyze whether the performance of LLMs plateaus after a given number of in-context examples or not.

In Appendix R we analyze the performance of LLMs whose backbone architecture is different from Transformers.

## B   Related Work (Expanded)

The in-context learning capability of large language models has garnered significant attention Brown et al. (2020). How this capability emerges during a standard next-token prediction pretraining and how it operates is still up for debate. A substantial body of research is dedicated to exploring the parallels between in-context learning mechanisms and traditional algorithms like gradient descent (Akyürek et al., 2023; von Oswald et al., 2022; Dai et al., 2023; Ahn et al., 2023; Cheng et al., 2023; Mahankali et al., 2024; Vladymyrov et al., 2024). For example, Akyürek et al. (2023) and von Oswald et al. (2022) prove that transformers could theoretically implement gradient descent. Bai et al. (2023) shows that the transformer architecture can implement more complex in-context learning procedures, involving algorithm selection. Cheng et al. (2023) argue that non-linear transformers learn to implement gradient descent in function spaces. von Oswald et al. (2023) suggests that performance of transformer-based models may be due to an architectural bias towards mesa-optimizaiton. Nonetheless, the extent to which pre-trained transformers actually implement gradient descent when given in-context examples remains a topic of debate (Natan et al., 2023; Shen et al., 2023).

Other lines of work investigate the convergence of in-context learning (Wies et al., 2023; Huang et al., 2023). Li et al. (2024) analyzes the training dynamics of transformers with nonlinear attention and nonlinear MLP, expanding upon previous work which considered simpler transformer-based architectures (Huang et al., 2023; Tian et al., 2023). However, for natural language tasks such as sentiment analysis, it is unclear how much learning occurs with in-context examples (Min et al., 2022; Pan et al., 2023; Kossen et al., 2024). For example,

Min et al. (2022) shows that GPT-3 retains a strong performance even when the labels of the in-context exemplars are random. On the other hand, recent work (Hendel et al., 2023; Liu et al., 2023) investigated how in-context learning creates task vectors, which can then be applied to produce the output.

Another question investigated in recent work is where does the in-context learning (ICL) emerges from (Chan et al., 2022; Xie et al., 2022; Han et al., 2023). For example, Chan et al. (2022) shows that in-context learning appears when the training data has particular properties. Xie et al. (2022) analyzes in-context learning through a small scale synthetic dataset (GINC). Han et al. (2023) identified a subset of the pre-training data that supports in-context learning, showing how continuing pretraining on this subset increases the model's ICL abilities.

Another line of research, which is close to our work, is that of investigating what types of "functions" can be learned through in-context learning (Garg et al., 2022; Zhang et al., 2023; Xing et al., 2024). Notably, all these works do not use pre-trained LLMs, but specifically train a transformer for the task. Garg et al. (2022) shows empirically that standard transformers can be trained from scratch to perform in-context learning of linear functions. Guo et al. (2024) investigates more complex function classes. Wei et al. (2023) shows that larger language models are able to overcome their semantic priors when shown input-label mappings. Zhang et al. (2023) train transformers with a single linear self-attention layer to in-context learn linear regression tasks, showing that transformers are capable of obtaining a performance competitive with the best linear predictor. Bhattamishra et al. (2024) experiment with training various models to in-context learn boolean functions. Although not the main focus of their work, they also experiment with pre-trained models such as Llama 2 and GPT-4, showing that they obtain a performance similar to nearest-neighbor baselines for boolean functions. A post on the AI Alignment Forum (Pesut) also explored the types of models that GPT-3 can fit in-context. While their focus was primarily on classification, they did examine regression scenarios as well.

While in this work we focus on in-context learning, there are other previous works that designed specific neural network modules (or cells) for arithmetic computations (Trask et al., 2018; Heim et al., 2020; Mistry et al., 2022).

Different from previous work, we investigate how pre-trained models, such as GPT-4 or Claude 3, *without any gradient updates*, can learn various linear and non-linear function classes when given examples in-context and thoroughly compare them against multiple traditional supervised methods (Ruppert, 2004) such as Gradient Boosting (Schapire, 1989; Friedman, 2001) or Random Forests (Breiman, 2001).

## C  Datasets

We provide the formulas for all datasets used below. We set the noise to 0 for all datasets.

### C.1  Linear Regression

In order to generate the linear regression datasets, we use the function make_regression, available in sklearn (Pedregosa et al., 2011).

### C.2  Friedman # 1

$$f(x) = 10 \cdot sin(x_0 x_1 \pi) + 20(x_2 - 0.5)^2 + 10x_3 + 5x_4 + \epsilon \cdot \mathcal{N}(0, 1)$$

Where $x_0, x_1, x_2, x_3, x_4 \sim U(0, 1)$

### C.3  Friedman # 2

$$f(x) = \sqrt{x_1^2 + \left(x_2 \cdot x_3 - \frac{1}{x_2 \cdot x_4}\right)^2} + \epsilon \cdot \mathcal{N}(0, 1)$$

Where

$$x_1 \sim \mathcal{U}(0, 100)$$
$$x_2 \sim \mathcal{U}(40\pi, 560\pi)$$
$$x_3 \sim \mathcal{U}(0, 1)$$
$$x_4 \sim \mathcal{U}(1, 11)$$

### C.4 Friedman # 3

$$f(x) = arctan\left(\frac{x_1 x_2 - \frac{1}{x_1 x_3}}{x_0}\right) + \epsilon \cdot \mathcal{N}(0, 1).$$

Where

$$x_1 \sim \mathcal{U}(0, 100)$$
$$x_2 \sim \mathcal{U}(40\pi, 560\pi)$$
$$x_3 \sim \mathcal{U}(0, 1)$$
$$x_4 \sim \mathcal{U}(1, 11)$$

### C.5 Original # 1

$$y = x + 10sin(\frac{5\pi x}{100}) + 10cos(\frac{6\pi x}{100}) \tag{4}$$

Where

$$x \sim \mathcal{U}(0, 100) \tag{5}$$

### C.6 Original # 2

$$y = (x_1^4 + (x_2 \cdot x_3 - \frac{2}{\sqrt{x_2} \cdot \sqrt{x_4}})^2)^{\frac{3}{4}}$$

Where

$$x_0 \sim \mathcal{U}(0, 3)$$
$$x_1 \sim \mathcal{U}(4\pi, 56\pi)$$
$$x_2 \sim \mathcal{U}(0, 2)$$
$$x_3 \sim \mathcal{U}(1, 11)$$

### C.7 Original # 3

$$y = e^{x_0} + \frac{x_1 \cdot x_2}{\sqrt{x_3}} + (x_0 \cdot x_3)^{\frac{3}{2}}$$

Where

$$x_0 \sim \mathcal{U}(1, 3)$$
$$x_1 \sim \mathcal{U}(1, 10)$$
$$x_2 \sim \mathcal{U}(0, 10)$$
$$x_3 \sim \mathcal{U}(1, 20)$$

### C.8   Original # 4

$$y = \frac{x_1}{10} \cdot sin(x_0) + \frac{x_0}{10} \cdot cos(x_1) + \frac{\sqrt{x_0}log(x_1)}{\sqrt{x_1}log(x_0)}$$

Where

$$x_0, x_1 \sim \mathcal{U}(2, 100)$$

### C.9   Original # 5

$$y = 100 * max(softmax(\frac{x}{10}))$$

Where

$$x_0, x_1, x_2 \sim \mathcal{U}(-25, 25)$$

### C.10   Neural Network Induced

For the random datasets induced by neural networks, we randomly initialize a neural network and create a dataset by feeding random data to it. The dataset Simple Random NN 1 was created by using a neural network with one hidden layer with ReLU non-linearities. The dataset Transformer 1 was created by a randomly initialized Transformer encoder block.

### C.11   Non-Numerical Regression

We provide the code to generate the non-numerical regression datasets in the Listing 1. Essentially, we assign a random number (0 to 26) to each lowercase letter. Then we sample a weight vector. The expected output is generated by doing a dot product between the underlying assigned value of each character and the generated weight vector.

Listing 1: The python code to generate non-numerical regression datasets

```
1  import random
2  import numpy as np
3  import string
4
5  max_num_vars = 5
6  n_samples = 51
7
8  def get_character_regression(random_state=1):
9      r = random.Random(random_state)
10     alphabet = list(string.ascii_lowercase)
11     shuffled = r.sample(alphabet, 26)
12     l2i = [(c, i) for c, i in enumerate(shuffled)]
13     gen = np.random.RandomState(random_state)
14
15     letters = r.sample(l2i, max_num_vars)
16     sample = r.choices(letters, k=max_num_vars * n_samples)
17     sample = np.array(sample)[:, 1].reshape(-1, max_num_vars)
18
19     weight_vec = 10 * gen.uniform(size=(max_num_vars))
```

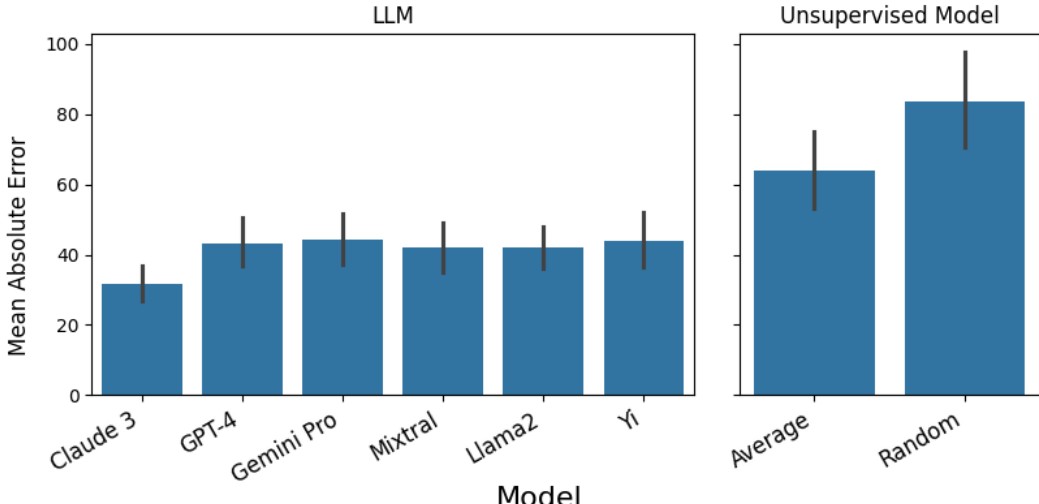

Figure 8: Performance Comparison between LLMs and unsupervised baselines on a non-numeric regression dataset. LLMs are outperforming our unsupervised heuristics even in this regime.

```
20
21      i2l = dict(letters)
22      l2i = {v: k for k, v in i2l.items()}
23
24      y = [weight_vec@np.array([l2i[c] for c in x]) for x in sample]
```

### C.12 Real-World Datasets

We also experimented with several real-world datasets: (i) Liver Disorders (UCI, 1990), (ii) Real Estate Valuation (Yeh, 2018), (iii) Diabetes (Efron et al., 2004), (iv) Servo (Ulrich, 1993), (v) Movies (Ahmed, 2017). The corresponding results are presented in Section I.

## D  Models

In the following, we provide additional details of the models we used for our main experiments and how we used them. We used three different types of models, as follows: (a) Large Language Models, (b) Traditional Supervised Methods, and (c) Heuristic-Based Unsupervised Methods. We describe them bellow.

### D.1  LLM

This section outlines the 12 Large Language Models (LLMs) featured in our main experiments, which include a mix of open-weights and private models. We also include additional models, such as the newest GPT-4 version (`gpt-4-20240409`), multiple Claude variants, and the most powerful model released by Cohere, Cohere Command R Plus. We tried with Cohere Command R and Cohere Command and observed their performance to be lower, albeit still the unsupervised baselines (except for Cohere Command).

In Table 2, we categorize the models by their names, availability of weights, and developers, dividing them into two distinct sections. The first section lists the models featured in the main paper's experiments (referenced in Sections 3, 4, and 5). The second section introduces additional models that were utilized for the extended analysis included in the Appendix.

| Model Name | Weights Availability | Developer |
|---|---|---|
| GPT-4 (Achiam et al., 2023) | Not available | OpenAI |
| Chat GPT (Ouyang et al., 2022) | Not available | OpenAI |
| Claude 3 Opus (Anthropic) | Not available | Anthropic |
| Claude 3 Sonnet (Anthropic) | Not available | Anthropic |
| Gemini Pro (Team et al., 2023) | Not available | Google |
| Mistral Medium | Not Available | Mistral |
| Mixtral Mixture of Experts 8x7B (Jiang et al., 2024) | Available | Mistral |
| Mistral 7B (Jiang et al., 2023) | Available | Mistral |
| Llama2 70B (Touvron et al., 2023) | Available | Meta |
| Code Llama2 70B (Rozière et al., 2023) | Available | Meta |
| Yi 34B (AI et al., 2024) | Available | 01.ai |
| DBRX[#] | Available | Databricks |
| GPT-4 (20240409) (Achiam et al., 2023) | Not available | OpenAI |
| GPT-3 Davinci (Brown et al., 2020) | Not available | OpenAI |
| GPT-3 Babbage (Brown et al., 2020) | Not available | OpenAI |
| Claude 3 Haiku (Anthropic) | Not available | Anthropic |
| Claude v2.1[†] | Not available | Anthropic |
| Claude v2.0[†] | Not available | Anthropic |
| Claude v1.2[‡] | Not available | Anthropic |
| Cohere Command R Plus[°] | Available | Cohere |
| Mixtral Mixture of Experts 8x22B | Available | Mistral |
| Falcon 40B Almazrouei et al. (2023) | Available | TII |
| Falcon 40B Instruct Almazrouei et al. (2023) | Available | TII |
| RWKV v4 14B Peng et al. (2023) | Available | Various[*] |
| StripedHyena Nous 7B Poli et al. (2023b) | Available | TogetherAI |

[†] https://www.anthropic.com/news/claude-2
[‡] https://www.anthropic.com/news/releasing-claude-instant-1-2
[#] https://www.databricks.com/blog/introducing-dbrx-new-state-art-open-llm
[°] https://cohere.com/command
[*] Developed collaboratively by numerous contributors across 29 different affiliations

Table 2: Details of the large language models (LLMs) used in our study, divided into two sections. The first section contains the LLMs used for the experiments from the main body of the paper, while the second section includes additional models explored in the extended results presented in the Appendix.

We list in Table 3 the models we used through OpenAI, together with their corresponding model code.[8]

| Model Name | API Model Code |
|---|---|
| GPT-4 | gpt-4-0125-preview |
| Chat GPT | gpt-3.5-turbo-1106 |
| GPT-4 (20240409) | gpt-4-turbo-2024-04-09 |
| GPT-3 Davinci | davinci-002 |
| GPT-3 Babbage | babbage-002 |

Table 3: The specific model used for each model family through the OpenAI API.

---

[8]https://openai.com/

We list in Table 4 the models we used through OpenRouter, together with their corresponding model code.[9]

| Model Name | API Model Code |
|---|---|
| Claude 3 Opus | `anthropic/claude-3-opus` |
| Claude 3 Sonnet | `anthropic/claude-3-sonnet` |
| Gemini Pro | `google/gemini-pro` |
| Mistral Medium | `mistralai/mistral-medium` |
| Claude 3 Haiku | `anthropic/claude-3-haiku` |
| Claude v2.1 | `anthropic/claude-2.1` |
| Claude v2.0 | `anthropic/claude-2.0` |
| Claude v1.2 | `anthropic/claude-1.2` |
| Cohere Command R Plus | `cohere/command-r-plus` |
| StripedHyena Nous 7B | `togethercomputer/stripedhyena-nous-7b` |

Table 4: The specific model used for each model family through the OpenRouter API.

We list in Table 5 the models we used through DeepInfra, together with their corresponding model code.[10]

| Model Name | API Model Code |
|---|---|
| Mixtral Mixture of Experts 8x7B | `mistralai/Mixtral-8x7B-Instruct-v0.1` |
| Mistral 7B | `mistralai/Mistral-7B-Instruct-v0.1` |
| Llama 70B | `meta-llama/Llama-2-70b-chat-hf` |
| Code Llama 70B | `codellama/CodeLlama-70b-Instruct-hf` |
| Yi 34B | `01-ai/Yi-34B-Chat` |
| Mistral 7B v2 | `mistralai/Mistral-7B-Instruct-v0.2` |

Table 5: The specific model used for each model family through the DeepInfra API.

We list in Table 6 the models we used through Fireworks, together with their corresponding model code.[11]

| Model Name | API Model Code |
|---|---|
| DBRX | `accounts/fireworks/models/dbrx-instruct` |
| Mixtral 8x22B | `accounts/fireworks/models/mixtral-8x22b` |

Table 6: The specific model used for each model family through the Fireworks API.

### D.1.1 Prompt

We show the prompt we used in Figure 9. Importantly, we used the same prompt for all large language models. We did not tune the prompt.

We encountered cases where the large language model would not produce a valid output. For instance, some models would occasionally output an empty string (i.e., ""). In the following, we detail the way we handled them across the experiments we showed in this paper.

---

[9] https://openrouter.ai
[10] https://deepinfra.com
[11] https://fireworks.ai/

> The task is to provide your best estimate for "Output". Please provide that and only that, without any additional text.
>
>
> Feature 0: -2.06
> Output: -81.93
>
> Feature 0: -0.64
> Output: -25.33
>
> Feature 0: 1.62
> Output:

Figure 9: The prompt we use for all LLMs. Concretely, we use an initial instruction to prevent the models from explaining their prediciton, something which we observed to happen for some models (e.g., Claude 3 Opus). Then, we give each input-output pair and finally ask the model to predict the value corresponding to the test input.

For the experiments performed with 100 random seeds and 50 random input-output tuples (i.e., $\mathbb{D}_{50}$), we simply skip the invalid generations. This is not problematic because it rarely occurs. For example, it has never occurred for the any of the experiments showed in Section 3 and Section 4. Moreover, given that we run each experiment with 100 random seeds, skipping the invalid generations will still leave us with enough samples to estimate the statistics. For example, out of the results presented in Section 3 and 4, the largest number of invalid generations for a given dataset was 11, by Mistral Medium on `Regression NI 1/2`. Claude 3 Opus produced only 2 invalid generations, for `Friedman #2`, and GPT-4 did not produce any invalid generations. One exception is for the results shown in Appendix J, as we observed that Claude 2.0 and Claude 2.1 produced invalid generations very often. For example, Claude 2.1 generated invalid generations for the dataset `Regression 1/1` in 71% of the cases. The reason for invalid generations was usually because Claude 2.1 refused to give "potentially misleading numerical estimates without proper context". A second exception is for Striped Hyena Nous 7B (Appendix R). We add a cross ($\times$) to the cell in the rank heatmap for every model-dataset pair where the number of valid generations is under 20.

For the experiments where we investigated how the performance of the models scale with the number of examples, we average 3 random runs for each dataset size. In this set of experiments, only Llama 70B generated invalid outputs a total of 3 times, for `Original #2` and `Friedman #2`. We skip the random runs with invalid generations.

### D.2 Traditional Supervised Models

We use a total of 11 traditional supervised methods, resulting in over 20 different configurations. Specifically, we used the following models:

1. Linear Regression: We used 4 variants of Linear Regression: (i) standard linear regression (Linear Regression), (ii) ridge (Ridge), (iii) lasso (Lasso), and (iv) Linear Regression with Polynomial Features (Linear Regression + Poly), where we used polynomial features of degree 2

2. Multi-Layer Perceptron: We used 6 variants of multi-layer preceptrons: 3 with different widths (MLP Wide 1, MLP Wide 2, MLP Wide 3) and 3 with different depths (MLP Deep 1, MLP Deep 2, MLP Deep 3)

3. Random Forest

4. Bagging

5. Gradient Boosting

6. AdaBoost

7. SVM: We used both a single SVM and an SVM paired with a Scaler (SVM + Scaler)

8. KNN: We used multiple variants of KNN, where we vary the number of neighbors, the type of distance used, and the power parameter for the Minkowski metric; We distinguish between them with a `v{index}`

9. Kernel Ridge

10. Spline

We used the `sklearn` implementation for each model.[12] Similar to the LLM case, we do not tune any hyperparameters. We use the default hyperparameters available in sklearn. We remark that these supervised baselines are very strong, as (1) many of them are the results of algorithms specifically designed for regression (e.g., Spline), (2) all perform parameter updates, and (3) the default hyperparameters, as set in widely-used statistical packages, have been refined over time to offer a reliable and generally strong performance across a variety of scenarios.

### D.3 Unsupervised Models

We use three heuristic inspired unsupervised models:

1. **Average:** Predicts the next value, $y_{n+1}$, as the mean of all preceding outcomes: $y_{n+1} = \frac{1}{n} \sum_{i=1}^{n} y_i$

2. **Last:** Uses the most recent observation tuple $(x_n, y_n)$ for prediction, such that $y_{n+1} = y_n$

3. **Random:** Predicts $y_{n+1}$ by randomly selecting from the set of prior observations $\{y_1, \ldots, y_n\}$. The final prediction is thus $y_{n+1} = sample([y_1, \ldots, y_n])$

The goal of these unsupervised models is to better put the performance obtained by LLMs into perspective.

## E   Large Language Models Can Do Linear Regression (Expanded)

We expand the barplots shown in Figure 2 with more models and more datasets. In particular, we show in Figures 10, 11, 12, 13, 14, 15 the performance of the models on six datasets for linear regression. Specifically, we used the following datasets:

1. Regression 1/1, a linear regression with only 1 variable, which is informative

2. Regression 1/2, a linear regression with 2 variables, and only 1 informative variable

3. Regression 1/3, a linear regression with 3 variables, and only 1 informative variable

4. Regression 2/2, a linear regression with 2 variables, both which are informative

5. Regression 2/3, a linear regression with 3 variables, and only 2 informative variables

6. Regression 3/3, a linear regression with 3 variables, all which are informative

We included the corresponding rank heatmap in Figure 16.

We make the following observations. First, Claude 3 Opus performs among the best for the linear regression case where there is only one informative variable (`Regression 1/1`, `Regression 1/2`, `Regression 1/3`), ranking among top 3 best performing models. The performance drops when there are more informative variables.

Second, we remark that all large language models perform better than all the unsupervised methods on all datasets.

---

[12]We used `sklearn 1.4.1.post1`.

Third, we remark that the large language models display a good overall performance. For example, there are 4 LLMs (i.e., Claude 3 Opus, Claude 3 Sonnet, GPT-4, DBRX) which perform better than all 3 variants of KNN over all the datasets used.

Fourth, there are specific LLMs which **always** perform better than Gradient Boosting, such as Claude 3 Opus and GPT-4. We remark that DBRX and Code Llama 70B outperform Gradient Boosting in 4 out of 6 datasets. [13]

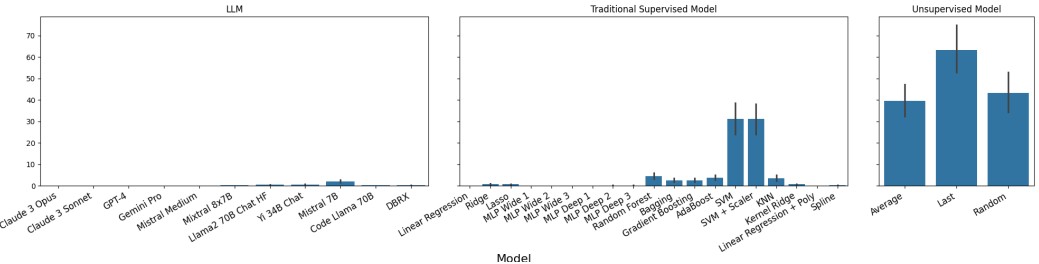

Figure 10: Regression 1/1

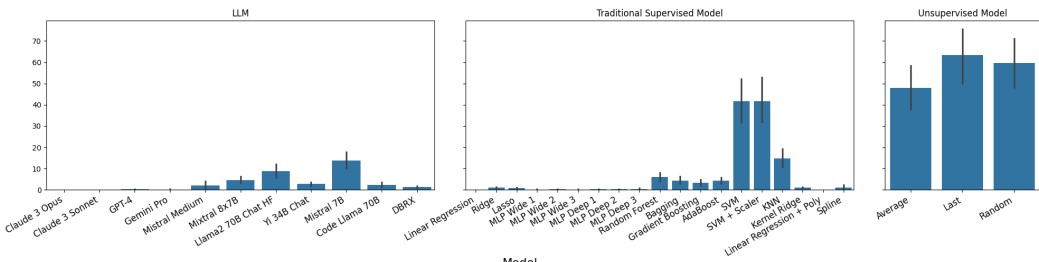

Figure 11: Regression 1/2

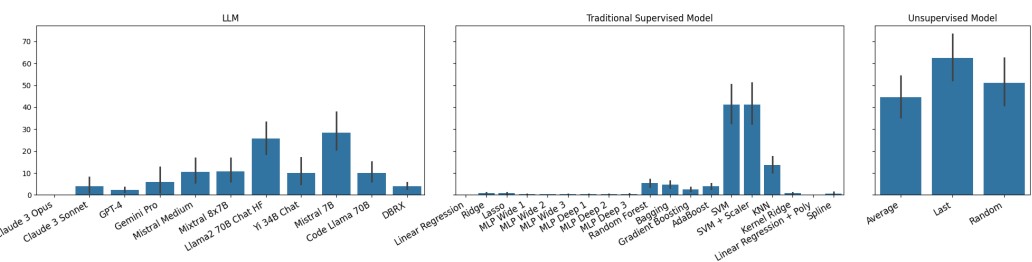

Figure 12: Regression 1/3

---

[13]In order to see how the performance of LLMs scale with more variables, we additionally experimented with Claude 3 Opus on Regression NI 1/10, 3/10, and 5/10 respectively. For example, in NI 5/10, Claude 3 Opus outperforms methods like Gradient Boosting or Random Forest but lags behind methods like Logistic Regression, which is expected given the nature of the datasets. For NI 3/10, Claude 3 Opus is outperformed by Random Forest and Gradient Boosting. Overall, we observed the performance to be similar to the performance on NI 2/3 from Figure 16.

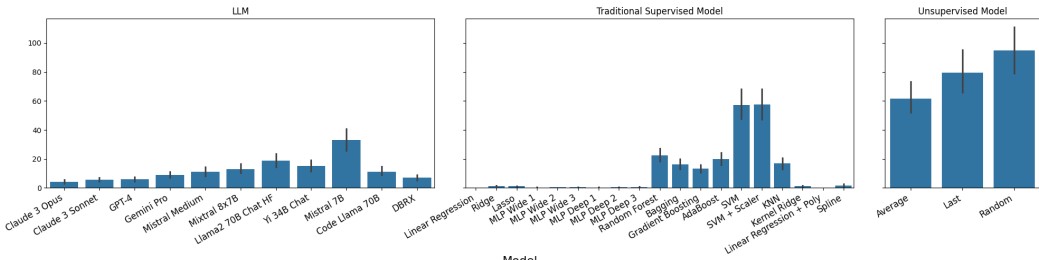

Figure 13: Regression 2/2

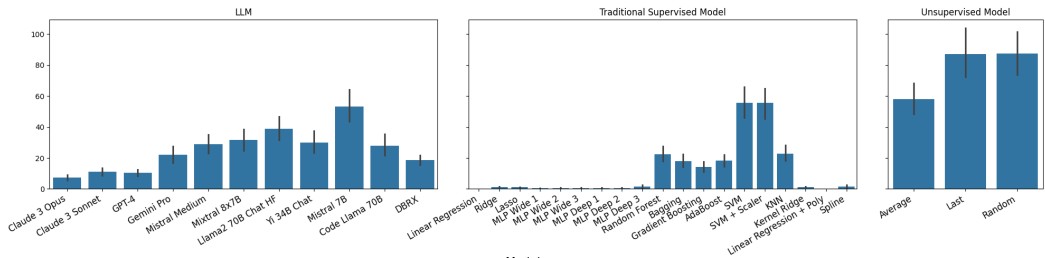

Figure 14: Regression 2/3

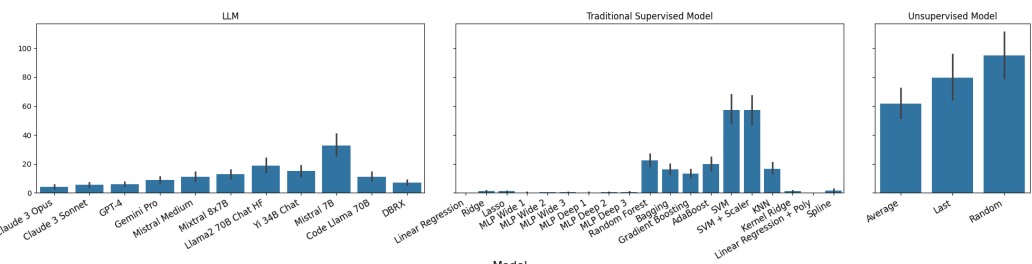

Figure 15: Regression 3/3

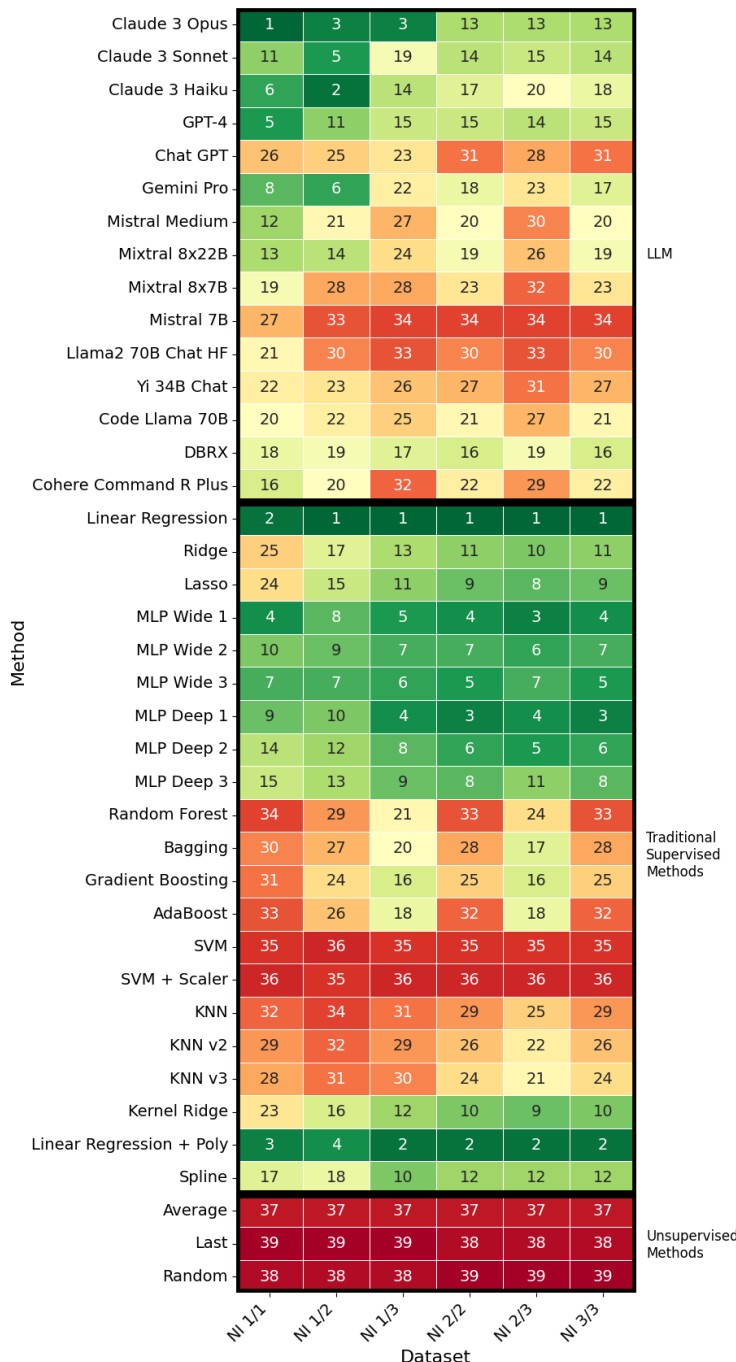

Figure 16: The rank of each model investigated on linear regression datasets. (best viewed in color)

# F  Large Language Models Can Do Non-Linear Regression (Expanded)

We expand the barplots shown in Figure 4 with more models and more datasets. In particular, we show in Figures 17, 18, 19, 20, 21, 22, 23, and 24 the performance of the models on the eight datasets for non-linear regression.

Additionally, we include in Figures 25 and 26 the performance of the models on datasets generated by randomly initialized neural networks, similar to the methodology of Garg et al. (2022). We can see that the performance of the LLMs decreases on the datasets created using the neural networks, although it (generally) remains above that of the unsupervised baselines.

Similar to Figure 6, we include the corresponding rankings in Figure 27.

We would like to remark that Claude 3 Opus obtains an average rank of 7.7, the best out of all models we have investigated. The next best is Gradient Boosting, with 8.1. The next best LLM is Claude 3 Sonnet, with 9.3, then GPT-4 with 12.7.

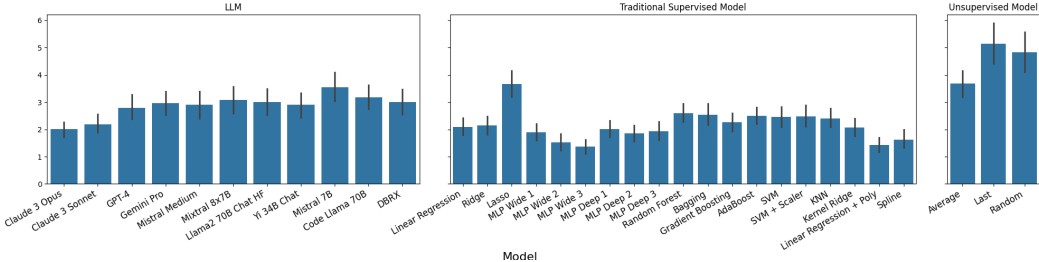

Figure 17: Extended results on the Friedman #1 dataset

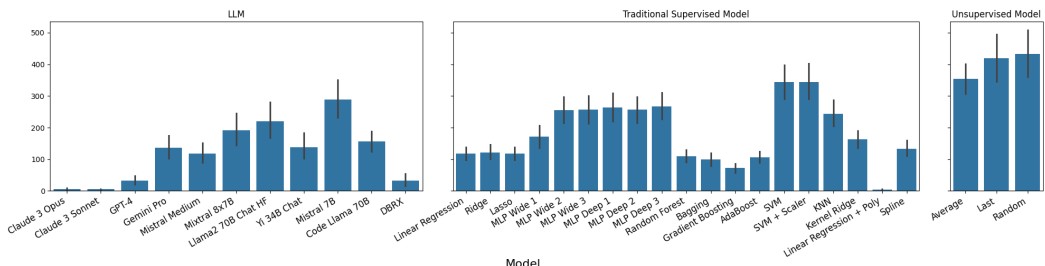

Figure 18: Extended results on the Friedman #2 dataset

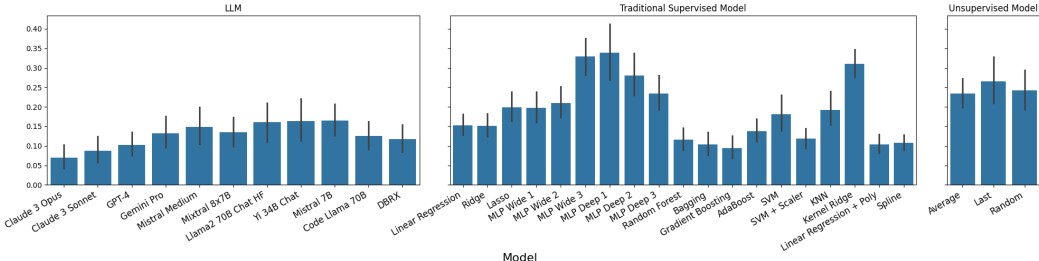

Figure 19: Extended results on the Friedman #3 dataset

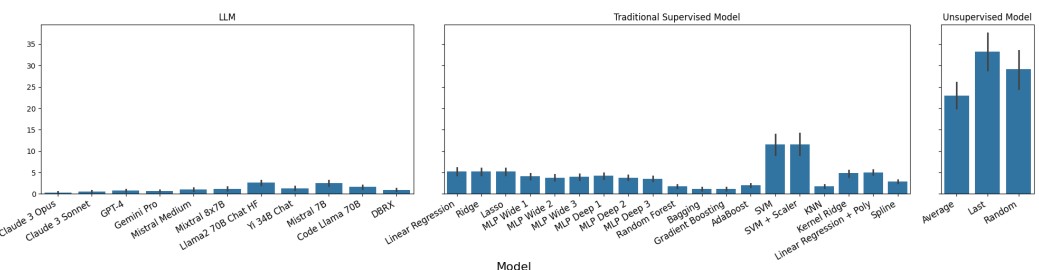

Figure 20: Extended results on the Original #1 dataset

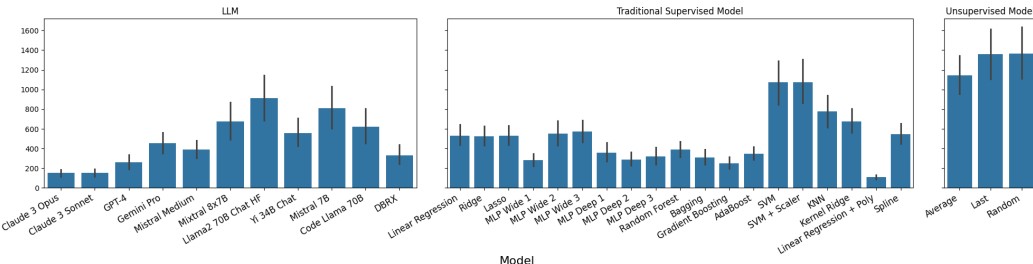

Figure 21: Extended results on the Original #2 dataset

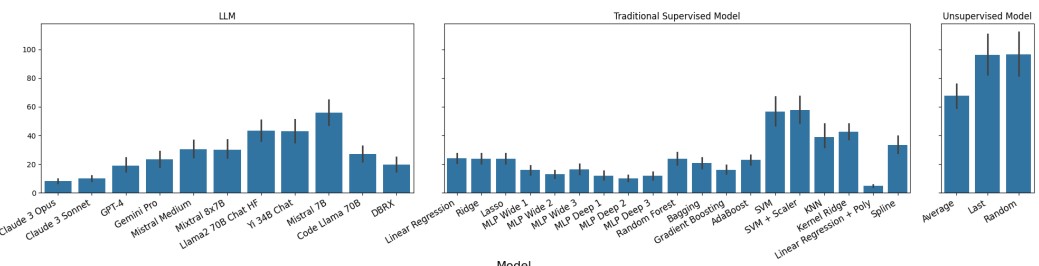

Figure 22: Extended results on the Original #3 dataset

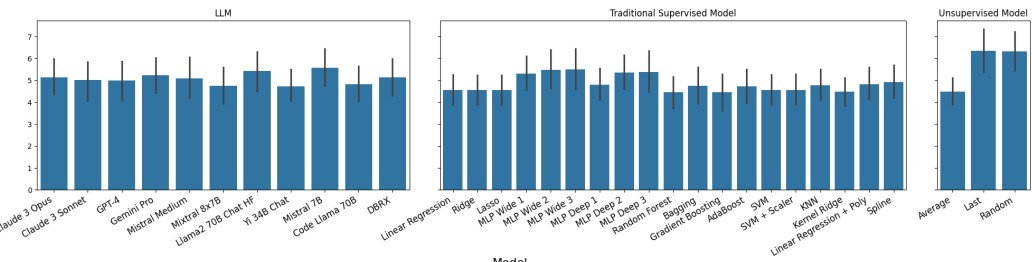

Figure 23: Extended results on the Original #4 dataset

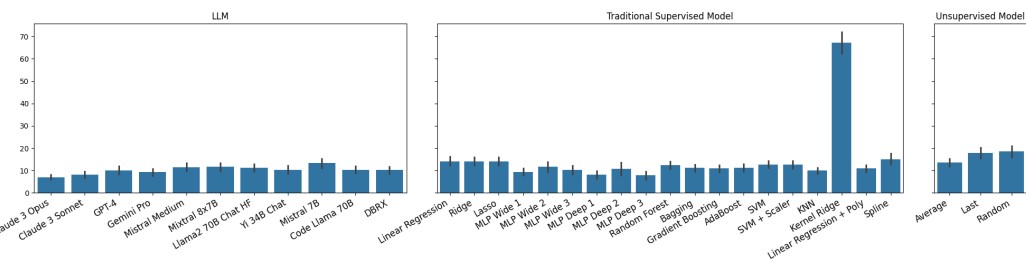

Figure 24: Extended results on the Original #5 dataset

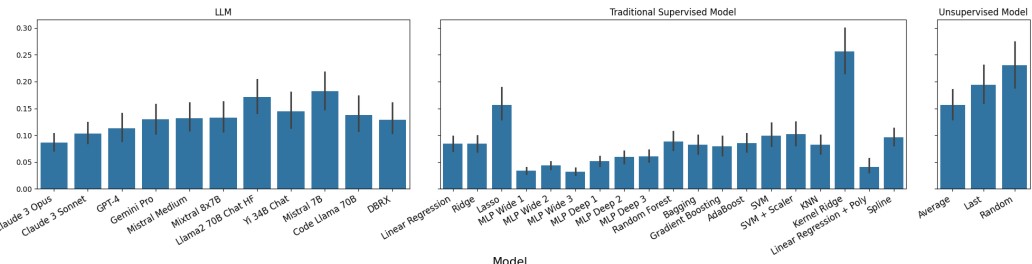

Figure 25: Results on a random regression dataset generated using a randomly initialized Neural Network

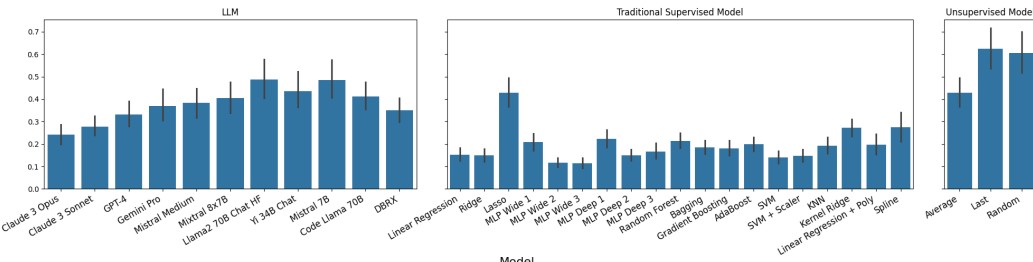

Figure 26: Results on a random regression dataset generated using a randomly initialized Transformer Encoder Block

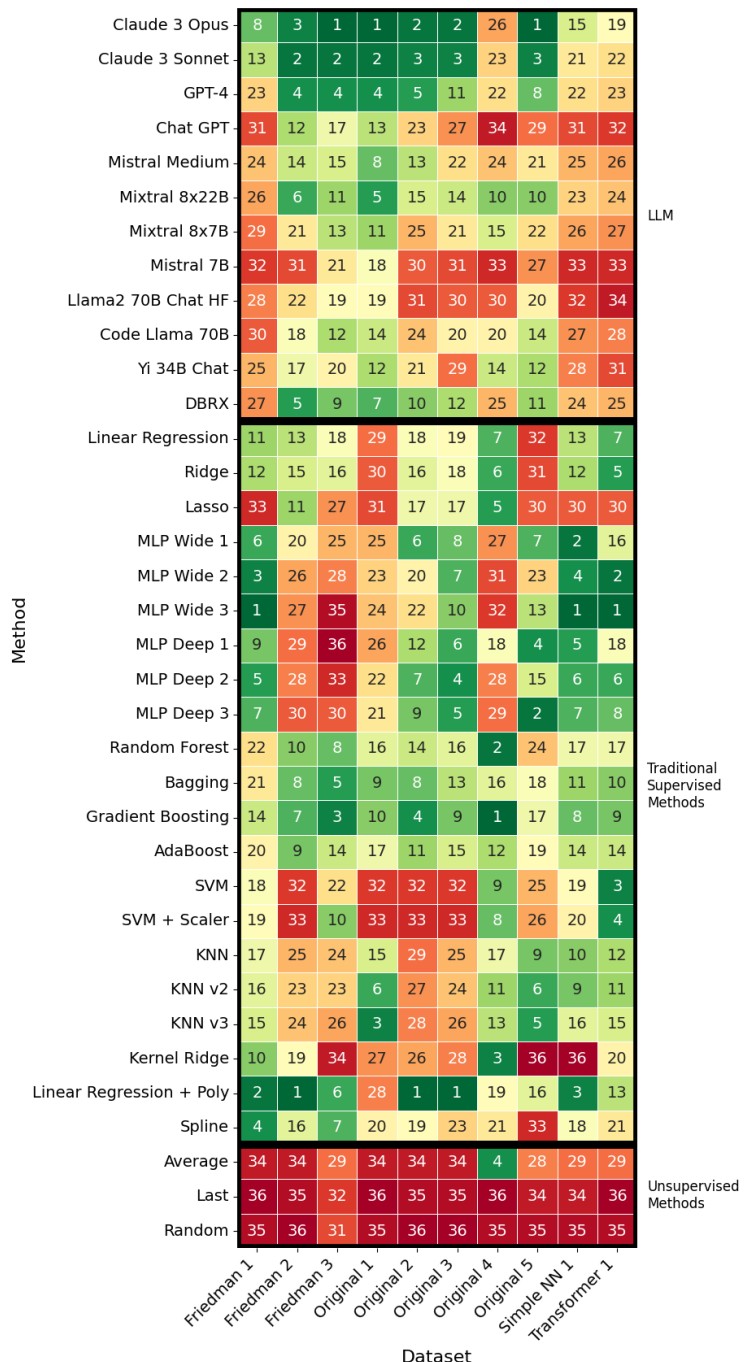

Figure 27: Rank of each model investigated on the non-linear regression datasets. (best viewed in color)

## G   Average Model Ranks

To provide a comprehensive overview of model performance across a diverse array of datasets, this section aggregates the average ranks obtained by each model. In this section we show the average ranks obtained by each model across: (1) all linear regression datasets (Linear), (2) all original benchmarking datasets introduced by us (Original), (3) all benchmarking datasets introduced by Friedman (Friedman), (4) all neural network induced datasets (NN), (5) all non-linear datasets (Non-Linear), (6) all datasets (Overall).

We show our results in Table 7. We divide the table into three blocks, separated by horizontal lines, corresponding to the results for (1) LLMs (e.g., GPT-4), (2) Traditional Supervised Methods (e.g., Gradient Boosting), and (3) Unsupervised Baselines (e.g., Average). We remark that LLMs obtain, overall, a strong performance. For example, Claude 3 Opus ranks second overall, outperforming methods such as Gradient Boosting, KNN, or multi-layer perceptrons. Overall, this strong performance is present in both private and open models. For example, DBRX and Mixtral 8x22B achieve average ranks that surpass conventional methods including AdaBoost, KNN, or Random Forests.

| Model | Linear | Original | Friedman | NN | Non-Linear | Overall |
|---|---|---|---|---|---|---|
| Claude 3 Opus | $7.67 \pm 5.89$ | $\mathbf{7.2 \pm 12.76}$ | $\underline{4.00 \pm 3.61}$ | $17.00 \pm 2.83$ | $\mathbf{8.2 \pm 9.99}$ | $8.00 \pm 8.45$ |
| Claude 3 Sonnet | $13.33 \pm 4.89$ | $\underline{7.4 \pm 10.41}$ | $\underline{5.67 \pm 6.35}$ | $21.5 \pm 0.71$ | $9.7 \pm 9.82$ | $11.06 \pm 8.31$ |
| Claude 3 Haiku | $13.33 \pm 7.69$ | $18.00 \pm 8.63$ | $19.00 \pm 10.44$ | $29.5 \pm 0.71$ | $20.6 \pm 8.92$ | $17.88 \pm 8.98$ |
| GPT-4 | $12.83 \pm 4.07$ | $11.4 \pm 8.02$ | $11.33 \pm 11.85$ | $22.5 \pm 0.71$ | $13.6 \pm 9.05$ | $13.31 \pm 7.4$ |
| GPT-4 (20240409) | $13.5 \pm 3.73$ | $12.00 \pm 9.41$ | $11.33 \pm 10.21$ | $25.00 \pm 1.41$ | $14.4 \pm 9.7$ | $14.06 \pm 7.83$ |
| Chat GPT | $29.5 \pm 3.33$ | $30.6 \pm 9.42$ | $24.67 \pm 12.9$ | $39.00 \pm 2.83$ | $30.5 \pm 10.23$ | $30.12 \pm 8.17$ |
| Davinci 002 | $26.17 \pm 6.27$ | $21.8 \pm 9.2$ | $23.33 \pm 5.03$ | $36.00 \pm 1.41$ | $25.1 \pm 8.77$ | $25.5 \pm 7.72$ |
| Babbage 002 | $40.17 \pm 1.83$ | $25.8 \pm 6.06$ | $39.67 \pm 0.58$ | $35.5 \pm 4.95$ | $31.9 \pm 7.92$ | $35.00 \pm 7.47$ |
| Gemini Pro | $16.5 \pm 7.4$ | $15.2 \pm 10.92$ | $21.33 \pm 7.09$ | $26.5 \pm 0.71$ | $19.3 \pm 9.3$ | $18.25 \pm 8.49$ |
| Mistral Medium | $23.17 \pm 6.71$ | $20.6 \pm 7.7$ | $20.33 \pm 5.13$ | $27.5 \pm 0.71$ | $21.9 \pm 6.4$ | $22.38 \pm 6.32$ |
| Mixtral 8x22B | $20.5 \pm 5.65$ | $12.4 \pm 4.51$ | $16.67 \pm 11.72$ | $23.5 \pm 0.71$ | $15.9 \pm 7.71$ | $17.62 \pm 7.18$ |
| Mixtral 8x7B | $27.17 \pm 4.92$ | $22.4 \pm 6.66$ | $26.00 \pm 9.54$ | $29.00 \pm 1.41$ | $24.8 \pm 6.91$ | $25.69 \pm 6.17$ |
| Mistral 7Bv2 | $35.00 \pm 2.53$ | $31.6 \pm 4.72$ | $26.33 \pm 7.64$ | $43.5 \pm 2.12$ | $32.4 \pm 7.96$ | $33.38 \pm 6.47$ |
| Mistral 7B | $37.33 \pm 2.66$ | $35.2 \pm 4.97$ | $36.67 \pm 7.57$ | $41.5 \pm 0.71$ | $36.9 \pm 5.49$ | $37.06 \pm 4.52$ |
| DBRX | $18.67 \pm 1.51$ | $15.00 \pm 8.0$ | $15.67 \pm 13.43$ | $25.00 \pm 0.0$ | $17.2 \pm 9.25$ | $17.75 \pm 7.25$ |
| Cohere Command R Plus | $25.00 \pm 6.16$ | $26.4 \pm 8.79$ | $24.67 \pm 10.07$ | $39.00 \pm 1.41$ | $28.4 \pm 9.43$ | $27.12 \pm 8.3$ |
| Llama2 70B Chat HF | $31.83 \pm 3.92$ | $32.8 \pm 7.22$ | $28.67 \pm 3.51$ | $40.5 \pm 3.54$ | $33.1 \pm 6.79$ | $32.62 \pm 5.76$ |
| Code Llama 70B | $24.5 \pm 2.88$ | $21.2 \pm 4.87$ | $23.67 \pm 11.59$ | $30.00 \pm 1.41$ | $23.7 \pm 7.27$ | $24.00 \pm 5.89$ |
| Yi 34B Chat | $28.00 \pm 2.76$ | $21.2 \pm 9.36$ | $25.00 \pm 3.46$ | $33.00 \pm 2.83$ | $24.7 \pm 8.04$ | $25.94 \pm 6.64$ |
| Gradient Boosting | $24.83 \pm 7.41$ | $9.6 \pm 7.33$ | $8.67 \pm 5.51$ | $8.5 \pm 0.71$ | $\underline{9.1 \pm 5.57}$ | $15.00 \pm 9.94$ |
| AdaBoost | $29.00 \pm 8.63$ | $17.6 \pm 4.62$ | $14.00 \pm 4.58$ | $14.00 \pm 0.0$ | $16.4 \pm 4.03$ | $21.12 \pm 8.62$ |
| Bagging | $27.00 \pm 6.48$ | $14.6 \pm 4.93$ | $12.00 \pm 8.19$ | $10.5 \pm 0.71$ | $13.00 \pm 5.37$ | $18.25 \pm 8.96$ |
| KNN | $32.67 \pm 5.05$ | $23.6 \pm 10.21$ | $27.33 \pm 8.96$ | $11.00 \pm 1.41$ | $22.2 \pm 10.11$ | $26.12 \pm 9.86$ |
| KNN v2 | $29.67 \pm 4.84$ | $18.2 \pm 12.87$ | $25.67 \pm 8.39$ | $10.00 \pm 1.41$ | $18.8 \pm 11.07$ | $22.88 \pm 10.53$ |
| KNN v3 | $28.67 \pm 5.28$ | $18.2 \pm 14.62$ | $27.33 \pm 10.79$ | $15.5 \pm 0.71$ | $20.4 \pm 12.04$ | $23.5 \pm 10.65$ |
| Linear Regression | $\mathbf{1.17 \pm 0.41}$ | $25.8 \pm 13.85$ | $16.00 \pm 5.57$ | $10.00 \pm 4.24$ | $19.7 \pm 11.84$ | $12.75 \pm 13.04$ |
| Lasso | $13.5 \pm 7.66$ | $24.8 \pm 14.86$ | $30.67 \pm 15.7$ | $33.5 \pm 0.71$ | $28.3 \pm 12.94$ | $22.75 \pm 13.22$ |
| Ridge | $15.33 \pm 7.28$ | $25.00 \pm 14.47$ | $16.33 \pm 4.04$ | $8.5 \pm 4.95$ | $19.1 \pm 12.1$ | $17.69 \pm 10.44$ |
| Linear Regression + Poly | $\underline{2.5 \pm 0.84}$ | $16.00 \pm 15.87$ | $\mathbf{3.00 \pm 2.65}$ | $8.00 \pm 7.07$ | $10.5 \pm 12.49$ | $\mathbf{7.5 \pm 10.48}$ |
| MLP Deep 1 | $5.67 \pm 3.39$ | $16.00 \pm 12.88$ | $31.33 \pm 20.11$ | $11.5 \pm 9.19$ | $19.7 \pm 15.51$ | $14.44 \pm 14.05$ |
| MLP Deep 2 | $8.83 \pm 4.17$ | $18.8 \pm 13.37$ | $28.67 \pm 20.98$ | $6.00 \pm 0.0$ | $19.2 \pm 15.68$ | $15.31 \pm 13.42$ |
| MLP Deep 3 | $11.00 \pm 3.35$ | $16.2 \pm 14.77$ | $29.00 \pm 19.16$ | $7.5 \pm 0.71$ | $18.3 \pm 15.66$ | $15.56 \pm 12.81$ |
| MLP Wide 1 | $4.67 \pm 1.75$ | $18.00 \pm 14.2$ | $21.67 \pm 14.29$ | $9.00 \pm 9.9$ | $17.3 \pm 12.95$ | $12.56 \pm 11.9$ |
| MLP Wide 2 | $7.83 \pm 1.83$ | $25.2 \pm 11.3$ | $25.00 \pm 19.16$ | $\underline{3.00 \pm 1.41}$ | $20.7 \pm 15.02$ | $15.88 \pm 13.34$ |
| MLP Wide 3 | $6.33 \pm 1.21$ | $24.4 \pm 11.72$ | $27.67 \pm 23.86$ | $\mathbf{1.00 \pm 0.0}$ | $20.7 \pm 17.25$ | $15.31 \pm 15.19$ |
| Random Forest | $31.5 \pm 7.04$ | $17.2 \pm 9.58$ | $14.33 \pm 6.81$ | $17.00 \pm 0.0$ | $16.3 \pm 7.27$ | $22.00 \pm 10.3$ |
| Kernel Ridge | $14.17 \pm 6.91$ | $30.00 \pm 15.72$ | $26.33 \pm 18.23$ | $34.00 \pm 19.8$ | $29.7 \pm 15.33$ | $23.88 \pm 14.74$ |
| SVM | $44.83 \pm 2.04$ | $32.8 \pm 14.57$ | $29.67 \pm 12.01$ | $11.00 \pm 11.31$ | $27.5 \pm 14.77$ | $34.00 \pm 14.4$ |
| SVM + Scaler | $45.5 \pm 1.76$ | $33.4 \pm 15.39$ | $24.33 \pm 16.65$ | $12.00 \pm 11.31$ | $26.4 \pm 15.99$ | $33.56 \pm 15.68$ |
| Spline | $14.00 \pm 3.95$ | $29.00 \pm 6.75$ | $10.33 \pm 7.77$ | $19.5 \pm 2.12$ | $21.5 \pm 10.38$ | $18.69 \pm 9.16$ |
| Average | $48.5 \pm 1.38$ | $34.2 \pm 17.43$ | $43.00 \pm 1.73$ | $33.5 \pm 0.71$ | $36.7 \pm 12.44$ | $41.12 \pm 11.32$ |
| Random | $51.17 \pm 0.98$ | $44.8 \pm 3.42$ | $46.67 \pm 3.21$ | $47.00 \pm 0.0$ | $45.8 \pm 2.94$ | $47.81 \pm 3.56$ |
| Last | $51.33 \pm 0.52$ | $45.00 \pm 4.3$ | $47.00 \pm 2.65$ | $47.00 \pm 1.41$ | $46.00 \pm 3.33$ | $48.00 \pm 3.72$ |

Table 7: We show the average rank of each model we investigated, across multiple types of datasets. We divide the table into three blocks, separated by horizontal lines, corresponding to the results for (1) large language models (e.g., GPT-4), (2) Traditional Supervised Methods (e.g., Gradient Boosting), and (3) Unsupervised Baselines (e.g., Average). Overall, the large language models (LLMs) obtain a strong performance. For example, Claude 3 Opus ranks second overall, outperforming very strong methods like Gradient Boosting or multi-layer perceptron. This strong performance is present in both private (e.g., Claude 3 Opus, GPT-4) and open-weights models (e.g., DBRX, Mixtral 8x22B).

## H How Fast Do Large Language Models Adapt? (Expanded)

We expand Table 1 to include more models. We show the corresponding results in Table 8.

Additionally, we include curve fit plots for models. To keep the number of plots to a manageable amount, we selected a subset of the models as follows. We selected Claude 3 Opus and GPT-4, as they are the flagship closed-source models. We selected Yi 34B Chat for the open-weights model. Lastly, we selected Gradient Boosting, Linear Regression, and Linear Regression + Poly. We present the corresponding plots in Figure 28 and 29. Since in these experiments we vary the number of in-context examples starting from 1, we could not include variants of KNN that uses more than one neigbhors for their prediction. We included KNN v4 which uses only one neighbor. Additionally, we included KNN v5, where we use a small number of neighbors when the amount of data is small, then gradually increase it. We can see that their performance is much worse than that of the LLMs, suggesting that the LLMs are doing something more than what KNN do.

| Model \Dataset | Friedman 1 | Friedman 2 | Friedman 3 | Original 1 | Original 2 | Regression NI 1/3 | Regression NI 2/2 |
|---|---|---|---|---|---|---|---|
| Claude 3 Opus | linear | sqrt | sqrt | log | sqrt | log | log |
| Claude 3 Sonnet | linear | log | linear | log | log | log | sqrt |
| GPT-4 | linear | sqrt | sqrt | log | sqrt | log | sqrt |
| Chat GPT | linear | linear | linear | sqrt | linear | sqrt | sqrt |
| Gemini Pro | linear | sqrt | linear | log | sqrt | sqrt | sqrt |
| Mistral Medium | linear | linear | linear | sqrt | sqrt | sqrt | sqrt |
| Llama2 70B Chat HF | linear | linear | linear | sqrt | linear | linear | sqrt |
| Code Llama 70B | linear | sqrt | linear | sqrt | sqrt | linear | sqrt |
| Mistral 7B | linear | linear | linear | sqrt | linear | linear | linear |
| Mixtral 8x7B | linear | linear | linear | sqrt | linear | linear | sqrt |
| Yi 34B Chat | linear | sqrt | linear | sqrt | sqrt | sqrt | sqrt |
| DBRX | linear | log | linear | log | sqrt | sqrt | sqrt |
| AdaBoost | linear | sqrt | linear | sqrt | sqrt | sqrt | sqrt |
| Bagging | linear | sqrt | linear | log | sqrt | sqrt | sqrt |
| Gradient Boosting | sqrt | sqrt | linear | log | sqrt | log | sqrt |
| Linear Regression | linear | linear | linear | linear | linear | log | log |
| Lasso | linear | linear | linear | linear | linear | log | log |
| Linear Regression + Poly | sqrt | log | log | linear | log | log | log |
| MLP Deep 1 | linear | linear | linear | linear | sqrt | log | log |
| MLP Deep 2 | linear | linear | linear | linear | sqrt | log | log |
| MLP Deep 3 | linear | linear | sqrt | linear | sqrt | log | log |
| MLP Wide 1 | linear | linear | linear | linear | sqrt | log | log |
| MLP Wide 2 | sqrt | linear | linear | linear | linear | log | log |
| MLP Wide 3 | sqrt | linear | linear | linear | linear | log | log |
| Random Forest | linear | sqrt | linear | sqrt | sqrt | sqrt | linear |
| Ridge | linear | sqrt | linear | linear | linear | log | log |
| SVM | linear | linear | linear | sqrt | linear | linear | linear |
| Kernel Ridge | linear | linear | linear | linear | linear | log | log |
| SVM + Scaler | linear | linear | linear | sqrt | linear | linear | linear |
| KNN v4 | linear | linear | linear | log | linear | sqrt | sqrt |
| KNN v5 | linear | linear | linear | log | linear | sqrt | sqrt |
| Average | linear | linear | linear | linear | linear | linear | linear |
| Last | linear | linear | linear | linear | linear | linear | linear |
| Random | linear | linear | linear | linear | linear | linear | linear |

Table 8: We show which curve-fit obtained the highest $R^2$ for multiple models and datasets, expanding on Table 1. The slower the growth of the function, the better (i.e., *log > sqrt > linear*). (best viewed in color)

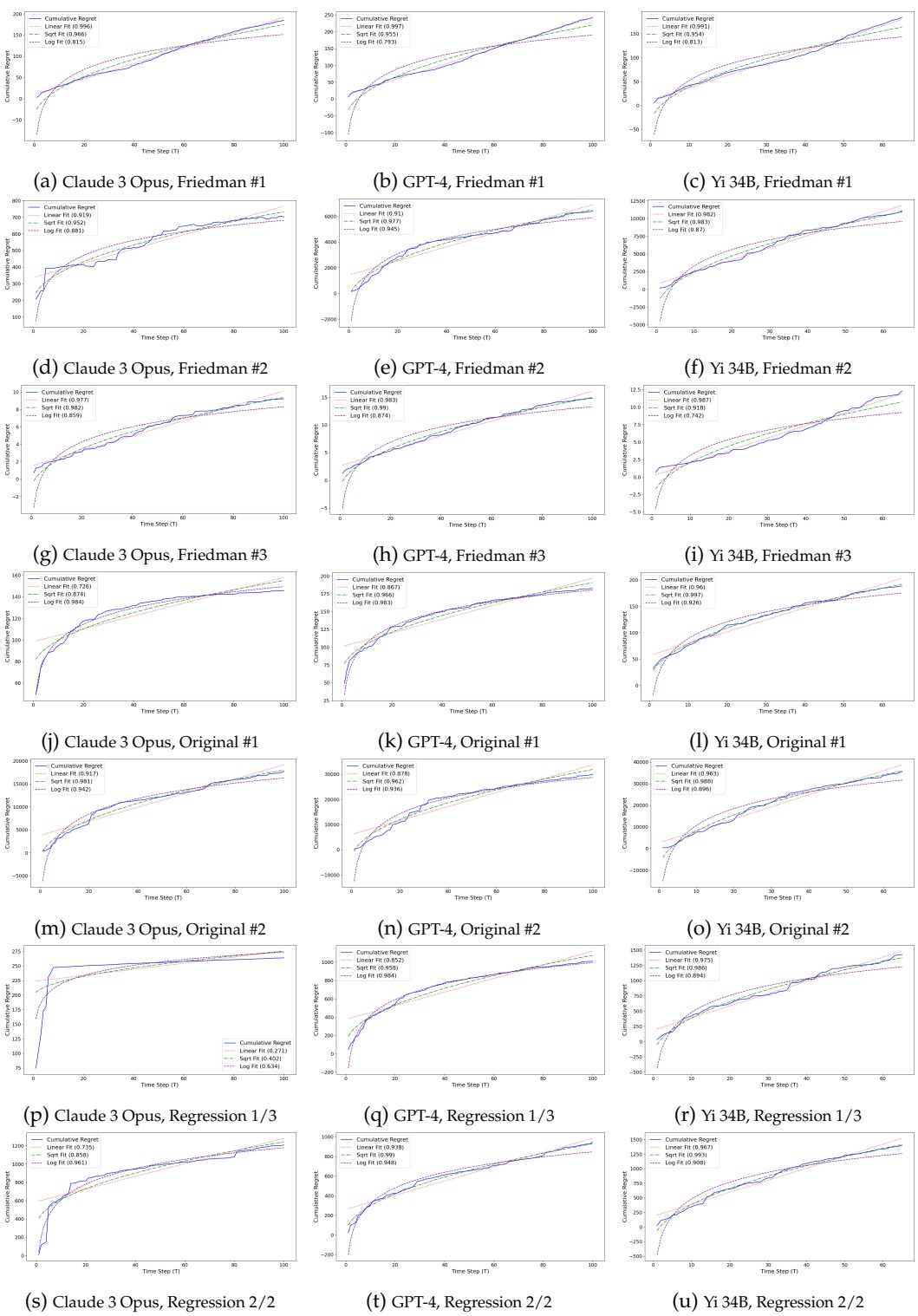

Figure 28: Curve Fits for Claude 3 Opus, GPT-4, and Yi 34B on seven (linear and non-linear) datasets.

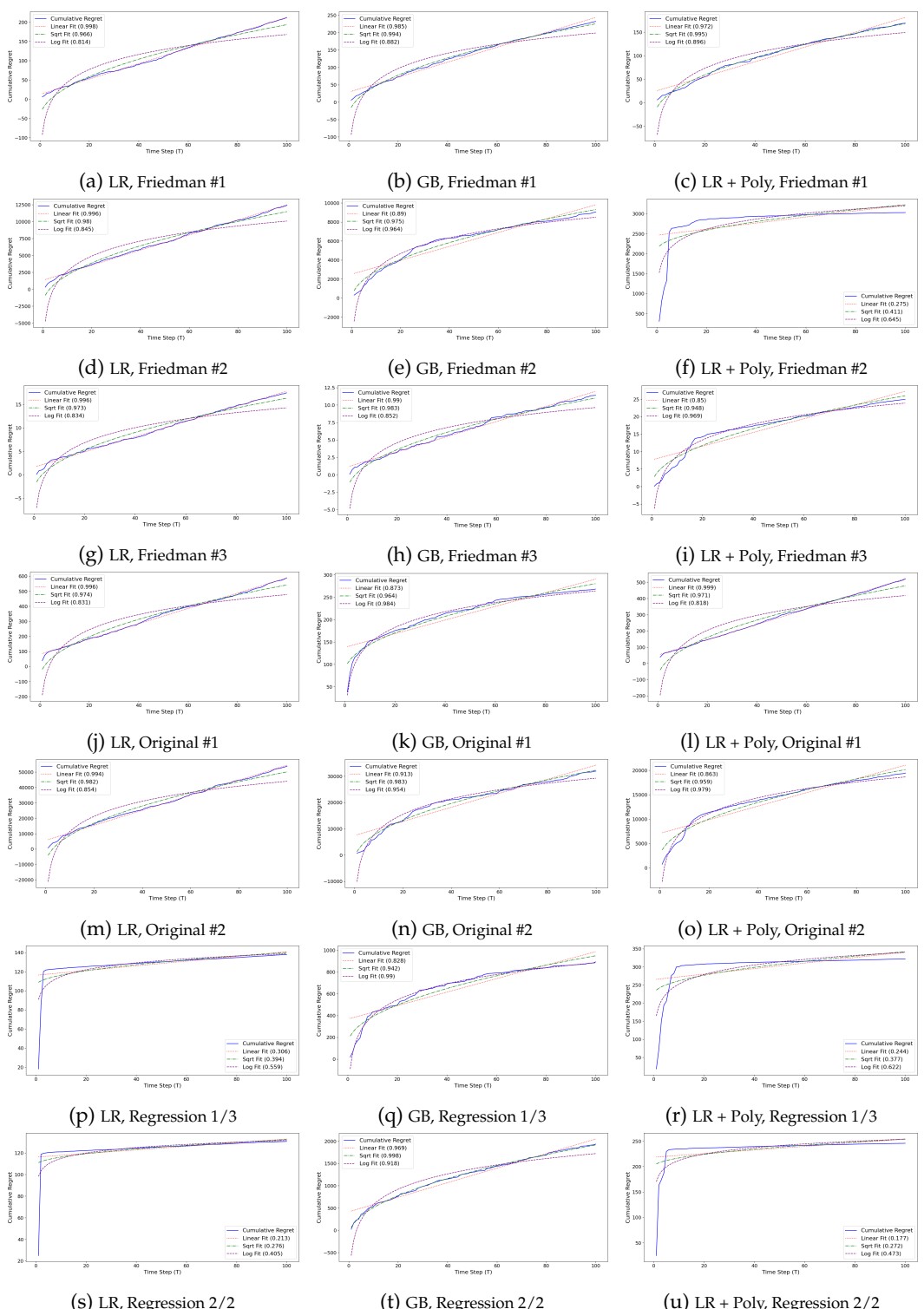

Figure 29: Curve Fits for Linear Regression (LR), Gradient Boosting (GB), and Linear Regression with Polynomial Features (LR + Poly) on seven (linear and non-linear) datasets.

# I   Performance on Real-World Datasets

In the following, we include the performance of GPT-4 and Claude 3 Opus on real-world datasets. Nevertheless, our aim in this work has not been to suggest that LLMs should replace (at least currently) traditional regression methods.[14] We evaluate on the following datasets:

1. Liver Disorders (UCI id=60 (UCI, 1990)
2. Real Estate Valuation (UCI id=477 (Yeh, 2018))
3. Diabetes (sklearn (Efron et al., 2004))[15]
4. Servo (UCI id=87 (Ulrich, 1993))
5. Movies (UCI id=424 (Ahmed, 2017))

We evaluated the following models:

- **LLMs**: GPT-4, Claude 3 Opus, Claude 3 Sonnet, Claude 3 Haiku;
- **Traditional Supervised Methods**: AdaBoost, Bagging, Gradient Boosting, KNN, KNN v2, KNN v3, Kernel Ridge, Lasso, Linear Regression, Linear Regression + Poly, MLP Deep 1, MLP Deep 2, MLP Deep 3, MLP Wide 1, MLP Wide 2, MLP Wide 3, Random Forest, Ridge, SVM, SVM + Scaler, Spline;
- **Unsupervised Methods**: Average, Last, Random.

We show our results in Tables 9, 10, 11, 12, and 13. In order to keep the tables easy to read, we selected: (i) two LLMs, Claude 3 Opus and GPT-4; (ii) three traditional supervised methods, Gradient Boosting, Random Forest, and Linear Regression with Polynomial Features (LR + Poly); and (iii) three unsupervised methods, Average, Last, and Random. We show both the mean absolute error (MAE) and the rank. The rank is shown against all the methods ran. For example, on Liver Disorders (Table 9), SVM ranks first.

| Model | MAE | Rank |
|---|---|---|
| GPT-4 | 2.55±2.49 | 6 |
| Gradient Boosting | 2.57±1.95 | 8 |
| Random Forest | 2.62±1.57 | 12 |
| Linear Regression + Poly | 2.67±1.61 | 19 |
| Average | 2.79±1.60 | 22 |
| Claude 3 Opus | 3.04±2.13 | 24 |
| Random | 3.23±2.71 | 26 |
| Last | 3.46±2.82 | 27 |

Table 9: Results on the Liver Disorders dataset, sorted by the MAE ($\downarrow$)

---

[14]Although we can foresee some interesting advantages for LLMs, such as allowing for natural language description of what each feature represents.

[15]https://scikit-learn.org/stable/modules/generated/sklearn.datasets.load_diabetes.html

| Model | MAE | Rank |
|---|---|---|
| Gradient Boosting | 4.30±4.91 | 1 |
| Claude 3 Opus | 5.08±5.35 | 4 |
| GPT-4 | 5.25±5.88 | 6 |
| Linear Regression + Poly | 5.38±5.78 | 7 |
| Random Forest | 5.72±5.32 | 10 |
| Average | 11.12±7.69 | 26 |
| Random | 14.39±10.05 | 27 |
| Last | 15.27±11.67 | 28 |

Table 10: Results on the Real Estate Valuation dataset, sorted by the MAE (↓)

| Model | MAE | Rank |
|---|---|---|
| Random Forest | 41.31±27.68 | 1 |
| Gradient Boosting | 47.95±28.36 | 16 |
| GPT-4 | 50.26±45.25 | 17 |
| Claude 3 Opus | 54.13±39.71 | 19 |
| Linear Regression + Poly | 55.84±52.87 | 20 |
| Average | 68.02±37.57 | 24 |
| Random | 94.34±63.17 | 26 |
| Last | 103.96±69.57 | 27 |

Table 11: Results on the Diabetes dataset, sorted by the MAE (↓)

| Model | MAE | Rank |
|---|---|---|
| Gradient Boosting | 0.25±0.29 | 2 |
| Claude 3 Opus | 0.27±0.46 | 4 |
| GPT-4 | 0.44±0.62 | 10 |
| Random Forest | 0.45±0.44 | 11 |
| Linear Regression + Poly | 0.62±0.38 | 13 |
| Average | 1.43±1.27 | 24 |
| Last | 1.61±1.56 | 26 |
| Random | 1.77±1.74 | 27 |

Table 12: Results on the Servo dataset, sorted by the MAE (↓)

| Model | MAE (1e+07) | Rank |
|---|---|---|
| Random Forest | 3.02±2.48 | 2 |
| Gradient Boosting | 3.03±2.51 | 3 |
| Claude 3 Opus | 4.51±6.20 | 17 |
| GPT-4 | 5.48±7.21 | 20 |
| Average | 6.20±4.85 | 21 |
| Linear Regression + Poly | 6.40±7.10 | 22 |
| Random | 8.17±10.9 | 23 |
| Last | 8.26±6.22 | 24 |

Table 13: Results on the Movies dataset (scaled by 1e+07), sorted by the MAE (↓)

## J Claude Performance

Following the (perhaps surprisingly) strong performance of the Claude family of large language models on various regression tasks (e.g., Figure 27, when averaging the ranks of each model over each dataset, Claude 3 Opus performs the best), we provide results with additional models from the Claude family, namely: Claude 1.2, Claude 2.0, Claude

2.1, and Claude 3 Haiku. We include a rank heatmap for all the models from the Claude family currently available in Figure 30. For comparison, we also included the corresponding performance of two strong models with open-weights: DBRX and Mixtral 8x7B. Claude 2.0 and Claude 2.1 were sometimes generating invalid outputs (e.g., "I apologize, upon reflection I do not feel comfortable providing output values without context. Could we have a constructive discussion about the meaning and implications of this exercise?"). Therefore, we omit those problematic configurations. For all the other cases, the average performance is the result of at least 20 runs.[16] We note that the performance of Claude 3 models is much better than that of older models.

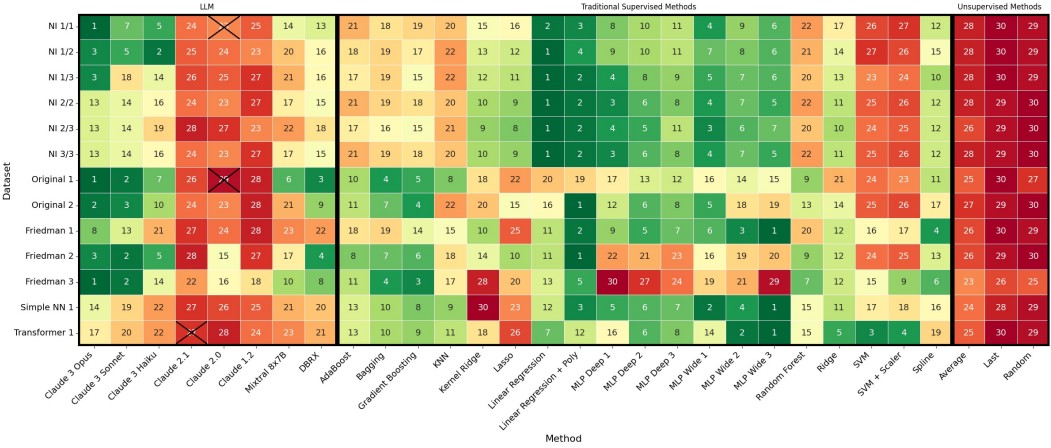

Figure 30: The rank of each model from the Claude family currently available. For comparison, we also included the ranks of two (strong) models with open-weights: DBRX and Mixtral 8x7B. (best viewed in color)

# K   Costs

We estimate the total cost for all our experiments to under $1200. We spent approximately $300 on OpenRouter. We spent approximately $600 on OpenAI. The cost for OpenAI is higher because we used it in our preliminary experiments. Additionally, the preliminary experiments used gpt-4, which is more expensive than gpt-4-0125-preview. We switched to gpt-4-0125-preview after we added the additional text to the prompt, instructing the model to only output their best estimate.[17]

# L   LLMs Justifying Their Prediction

Without adding the prefix instruction text (see prompt in Appendix D.1.1), which instructed the models to output **only** its best estimate, some LLMs (e.g., Claude 3 Opus) started to provide explanations, in an attempt to justify their prediction.[18] Analyzing these "explanations" revealed that there is a discrepancy between their explanation and their prediction. For example, for a non-linear regression problem, Claude 3 Opus suggests to train a Linear Regression model and gives the code to do so. Then it elaborates on how to use it for inference and gives the output. However, manually running the code suggested by the model results in a very different output. We provide some examples in Figures 31, 32, , 33, and 34. We describe each one below.

---

[16]These invalid outputs are specific to Claude 2.0 and Claude 2.1. For example, for Claude 3 Opus, there exist only 2 instances where it does not generate a valid output, out of a total of over 1000 runs.

[17]We did not need this additional instruction in our initial experiments. We added it when we expanded the number of LLMs used, as some of them (e.g., Claude 3) would provide justifications before giving the final output. All models use the same prompt.

[18]We hypothesize that this is because their system prompt instructs them to provide explanations.

In Figure 31, Claude 3 correctly identifies that the output is generated by multiplying the input with a constant. Then, it calculates the constant and gives the final output.

In Figure 32, we can see that Claude 3 first calculates the mean of each feature. Manually inspecting the true values, they are: Mean Feature 0: 51.3180, Mean Feature 1: 846.6326, Mean Feature 2: 0.4764, Mean Feature 3: 5.2438. We remark that the values given by Claude 3 are surprisingly close. Then, Mean Output: 374.04. Then, Claude 3 calculate the covariance between each feature and the output. These estimates are much worse. For example, Cov(Feature 0, Output) is actually 2516.3683, not 8958.8469. Next, Claude 3 gives the variance of each feature. These values are close. For example, Var(Feature 0) is 729.420751 and Claude 3 generates 729.9052. Then, the model calculates the coefficients. These calculations are close to their true value, except for b0. Lastly, the model gives the final formula to calculate the output, which is (according to the model): $227.4744 + 25.9730 * 42.54 + 1.2777 * 851.93 + 1648.5958 * 0.51 + 129.2408 * 6.26 = 436.5981$. However, this calculation is wrong. The output of that equation is 4070.70. However, we would like to remark that what the model generated (wrongly, from a mathematical point of view), is actually much closer to the true value of 434.54. In other words, the explanation offered by the model was not faithful.

In Figure 33, we can see that Claude 3 suggests that there is a strong linear relationship (please refer to Figure 5 for a plot of the data). Then, Claude 3 fits a linear regression $y = mx + b$ and gives the approximate values: $m = 0.9102$ and $b = 12.1615$. However, manually fitting a linear regression model on the corresponding data yields the following values $m = 0.97$ and $b = 2.94$. Then, Claude 3 calculates the final output. The calculation is correct, however it is far off from the true value of: 30.86. We would like to remark that instructing the model to give its best estimate without any additional information gives a much better prediction: 30.91.

In Figure 34, the solution generated by Claude 3 involves calculating the nearest neighbor. The problem in this approach is that the dataset given in-context contain only 50 examples, while the solution generated Claude involves taking the examples 50 and 54, which are non-existent.

All in all, we remark that the explanations provided by the model are not always faithful. We also remark that the predictions of the model in two cases: (i) when it outputs an explanation and (ii) when it does not output an explanation can vary.

To find the output for a given Feature 0 value, we need to determine the relationship between Feature 0 and the Output. Looking at the provided data points, it appears that the Output is approximately equal to Feature 0 multiplied by a constant value.

Let's calculate this constant value using a few data points:

For Feature 0 = -2.06, Output = -81.93
Constant = Output / Feature 0 = -81.93 / -2.06 $\approx$ 39.77

For Feature 0 = 1.46, Output = 58.14
Constant = Output / Feature 0 = 58.14 / 1.46 $\approx$ 39.82

The constant values are very close, so we can assume that the relationship is:
Output = Feature 0 $\cdot$ 39.8 (rounded to one decimal place)

Now, for Feature 0 = 1.62, we can calculate the Output:
Output = 1.62 $\cdot$ 39.8 $\approx$ 64.48

Therefore, when Feature 0 is 1.62, the Output should be approximately 64.48.

Figure 31: Claude 3 Opus justifying its output on Regression NI 1/1. In this dataset, the final output is constructed by multiplying the input feature with a constant value. Claude 3 correctly identified the constant. The answer is very close to the correct one (differences might appear due to rounding).

To predict the output value, I'll use a linear regression model with the given features. The model is:

Output = b0 + b1*Feature0 + b2*Feature1 + b3*Feature2 + b4*Feature3

First, I'll calculate the mean of each feature and the output:

Mean Feature 0: 51.7054
Mean Feature 1: 836.5202
Mean Feature 2: 0.4844
Mean Feature 3: 5.3154
Mean Output: 385.2352

Next, I'll calculate the covariance between each feature and the output:

Cov(Feature 0, Output): 18958.8469
Cov(Feature 1, Output): 262416.1385
Cov(Feature 2, Output): 138.6469
Cov(Feature 3, Output): 1117.1448

And the variance of each feature:

Var(Feature 0): 729.9052
Var(Feature 1): 205416.6328
Var(Feature 2): 0.0841
Var(Feature 3): 8.6469

The coefficients are:

b1 = Cov(Feature 0, Output) / Var(Feature 0) = 25.9730
b2 = Cov(Feature 1, Output) / Var(Feature 1) = 1.2777
b3 = Cov(Feature 2, Output) / Var(Feature 2) = 1648.5958
b4 = Cov(Feature 3, Output) / Var(Feature 3) = 129.2408

b0 = Mean(Output) - b1*Mean(Feature 0) - b2*Mean(Feature 1) - b3*Mean(Feature 2) - b4*Mean(Feature 3)
= -227.4744

So, the linear regression model is:

Output = -227.4744 + 25.9730*Feature0 + 1.2777*Feature1 + 1648.5958*Feature2 + 129.2408*Feature3

Plugging in the given feature values:

Output = -227.4744 + 25.9730*42.54 + 1.2777*851.93 + 1648.5958*0.51 + 129.2408*6.26
= 436.5981

Therefore, the predicted output value is 436.60.

Figure 32: Claude 3 Opus justifying its output on `Friedman #2`. Except for covariance and b0, the calculations made by Claude are close to their true values (e.g., Cov(Feature 0, Output) is 2516.3683, not 8958.8469). The model gives the final formula to calculate the output: $227.4744 + 25.9730 * 42.54 + 1.2777 * 851.93 + 1648.5958 * 0.51 + 129.2408 * 6.26 = 436.5981$. However, this calculation is wrong. The output of that equation is 4070.70. However, we would like to remark that what the model generated (wrongly, from a mathematical point of view), is actually much closer to the true value of 434.54. In other words, the explanation offered by the model was not faithful, even though the final prediction was good.

To find the relationship between the input Feature 0 and the Output, let's plot the data points and see if there's a clear pattern or trend.

Plotting the data, it appears there is a strong linear relationship between Feature 0 and Output. As Feature 0 increases, the Output also tends to increase in a fairly straight line.

To quantify this linear relationship, we can calculate the line of best fit using linear regression. The equation for a line is typically written as:

y = mx + b

where:
y is the Output value
m is the slope of the line
x is the Feature 0 value
b is the y-intercept (the Output value when Feature 0 is 0)

Using the given data points, the linear regression equation comes out to approximately:

Output = 0.9102 * Feature 0 + 12.1615

So for the given Feature 0 value of 31.34, plugging that into the equation:

Output = 0.9102 * 31.34 + 12.1615
= 28.53 + 12.1615
= 40.69

Therefore, based on the strong linear trend in the data, the predicted Output value corresponding to a Feature 0 value of 31.34 would be around 40.69.

Figure 33: Claude 3 Opus justifying its output on Original #1. In this dataset, the final output is constructed as follows: $f(x) = 10x + sin(5\pi x) + cos(6\pi x)$. The solution proceeds by fitting a linear regression and giving the approximate values. However, manually fitting a linear regression model yields the following values: $m = 0.97$ and $b = 2.94$. Then, the model proceeds by calculating the final output. The calculation is correct, but the final value is far off from the true value, which is 30.86. We would like to remark that instructing the model to give its best estimate without any additional information or explanation gives a much better prediction: 30.91.

To predict the output for the given input features, I'll calculate the average output for the two nearest neighbors in the training data.

Input: Feature 0: 0.7, Feature 1: -2.03

Calculating Euclidean distances to all training examples:

1. $\sqrt{(0.7 - (-1.06))^2 + (-2.03 - (-0.91))^2} = 2.27$
2. $\sqrt{(0.7 - 0.37)^2 + (-2.03 - 1.36)^2} = 3.45$
...
49. $\sqrt{(0.7 - (-0.38))^2 + (-2.03 - 0.07)^2} = 2.35$
50. $\sqrt{(0.7 - 1.17)^2 + (-2.03 - (-2.34))^2} = 0.62$
51. $\sqrt{(0.7 - (-0.46))^2 + (-2.03 - 0.35)^2} = 2.76$
52. $\sqrt{(0.7 - (-0.75))^2 + (-2.03 - 0.01)^2} = 2.64$
53. $\sqrt{(0.7 - 0.38)^2 + (-2.03 - (-0.22))^2} = 1.88$
54. $\sqrt{(0.7 - (-0.4))^2 + (-2.03 - (-1.38))^2} = 1.41$

The two nearest neighbors are examples 50 and 54, with distances 0.62 and 1.41.

Example 50 output: 31.76
Example 54 output: -49.65

Predicted output = (31.76 + (-49.65)) / 2 = -8.945

So for input features 0.7 and -2.03, the predicted output using 2-nearest neighbors is -8.945.

Figure 34: Claude 3 Opus justifying its output on `Regression NI 2/2`, a linear regression dataset with two variables, where both are informative. However, the dataset given as in-context exemplars to Claude contains only 50 data points, while the solution generated by Claude involves taking the examples 50 and 54. Those examples are non-existent.

## M  Beyond Numerical Regression

Our investigation has centered on conventional regression tasks characterized by inputs and outputs represented as numerical values. However, the performance on these tasks might be influenced by the quality of numerical token embeddings, which can serve as a confounding factor Razeghi et al. (2022). To address this and broaden our analysis, we shift our focus to datasets generated following the methodology outlined in Section 2.1.3. This allows us to evaluate the models' capabilities in contexts where inputs are symbolic rather than numerical.

Specifically, we define an input vocabulary $\mathcal{V}$, consisting of a small number of (random) symbols.[19] Each symbol is *randomly* assigned a number $\{1, \ldots, 26\}$. We map the symbols to a numerical value by sampling a weight vector $w \in \mathbb{R}^d$ and doing a dot product between it and the corresponding values of each symbol. We present our results in Figure 8. The large language model display, once again, a performance superior to that of unsupervised models.

## N  Effects of Rounding

Due to computational budgets and context limits,[20] we rounded both the input and the output to two decimals. To validate that our conclusions are not an artifact of the rounding mechanism, we re-ran GPT-4 on Friedman #2 and Friedman #3, rounding to five decimals. We selected GPT-4 because it obtained overall strong results and offers a good latency. We selected Friedman #2 and Friedman #3 because LLMs obtained generally good performance. We also ran the traditional supervised methods. We include the updated results in Figure 35. Comparing it with Figure 27, we can see that the performance of GPT-4 remains strong. This time it even outperforms Gradient Boosting on `Friedman #3` and ranks first. The performance on `Friedman #3` is only under Linear Regression + Poly, similar to Figure 27. All in all, the strong performance we observed is unlikely to be just an artifact of rounding.

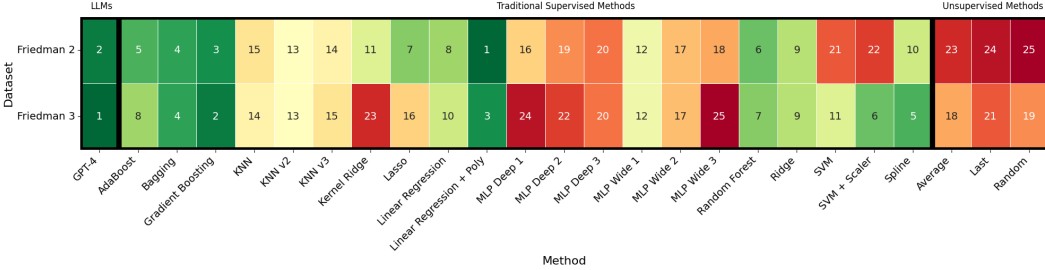

Figure 35: The ranks of the models when rounding to five decimals. The performance of GPT-4 remains strong, ranking above many traditional supervised methods. This is in line with our previous observations, where we rounded to two decimals. (best viewed in color)

## O  Is It Just Better KNN?

We can see from Appendix E and F that the performance of certain LLMs is almost always better than that of all three variants of KNN presented. For example, in the case of linear regression, Claude 3 Opus, Claude 3 Sonet and GPT-4 **always** perform better than the three variants of KNN we used. Similar, in the case of non-linear regression, only for `Simple NN 1` and `Transformer 1` is *any* of the KNN variants outperforming Claude 3 Opus. Moreover, from Table 8, which records the best curve fit over the cumulative regret, we can see that both variants of KNN perform worse than Claude 3 Opus or GPT-4.

---

[19]We used 5 symbols, in the form of letters (e.g., *a*)
[20]For example, Llama2 context size is only 4096.

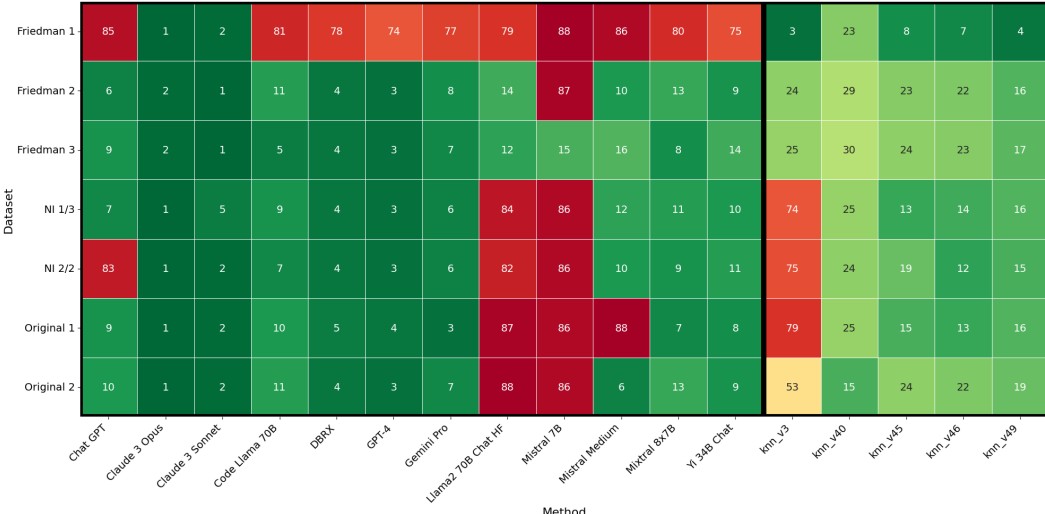

Figure 36: Comparison between the ranks of LLMs and the ranks of best KNN on each dataset. (best viewed in color)

To further investigate the extent to which what LLMs are internally implementing to do regression is a variant of KNN, we compare the performance of the LLMs against a total of 70 KNNs in a setting similar to that from Section 3 and 4: we randomly sample a dataset of size 50, which is given to both LLMs and KNNs and we ask them to predict the output corresponding to a testing data point. We repeat each experiment 100 times, with different random seeds. We describe what KNNs we considered in Listing 2.

Listing 2: The python code to create the KNN configurations

```
1  for n in [1, 2, 3, 5, 7, 9, 11]:
2      for w in ['uniform', 'distance']:
3          for p in [0.25, 0.5, 1, 1.5, 2]:
4              yield KNeighborsRegressor(n_neighbors=n, weights=w, p=p)
```

We summarize our results in Figure 36. To keep the plot comprehensible, we chose the best-performing KNN configuration for each dataset. However, the ranking considers the performance of all models. We draw the following conclusions.

First, we remark that except on `Friedman #1`, for **every other dataset**, the top 9 best performing models are all LLMs. In other words, both closed-source models (e.g., Claude 3 Opus, GPT-4) and open-weights models (e.g., DBRX, Mixtral) outperform all KNN models on all the datasets except `Friedman #1`. This suggests that the mechanism implemented by in-context learning might be something more complex than KNN.

Last, we remark that for `Friedman #1`, only Claude 3 Opus and Claude 3 Sonnet outperform the KNN variants. Moreover, Claude 3 Opus and Claude 3 Sonnet outperform **all** KNN variants we experimented with on **all** datasets.

## P  Could It Be Just Contamination?

The very large datasets that are used to train contemporary large language models (LLMs) raise concerns about potential contamination (Sainz et al., 2023; Golchin & Surdeanu, 2024). In our study, we have attempted to mitigate this as follows. First, we used many different random seeds. However, this does not nullify the risk that the LLM has seen similar data (e.g., data from `Friedman #1`, but with other random seeds). To mitigate this, we explored the performance of the models on regression functions of our own creation. This makes it unlikely that the model has seen data coming from the exact same function. For example,

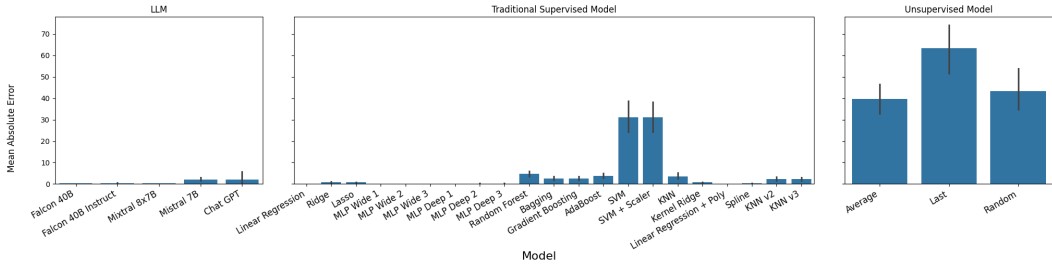

Figure 37: Falcon performance on the Regression NI 1/1 #2 dataset

across all our newly introduced datasets (e.g., `Original #1`, `Original #2`, `Original #3`, `Original #4`, `Original #5`), Claude 3 Opus obtains the highest average rank of 6.4. Second place is Clade 3 Sonnet, with an average rank of 6.8, then Gradient Boosting with 8.4. Furthermore, our empirical evidence of consistent high performance across a diverse array of LLMs.

To further analyze the data contamination issue, we perform two additional experiments. We provide results with Falcon, an LLM whose training data is publicly available. Second, we perform an experiment similar to the approach proposed in Golchin & Surdeanu (2024), where we compare the performance of LLMs with and without knowing the dataset where the data comes from.

### P.1 LLMs With Known Training Data

In this section we expand our analysis to include Falcon 40B and Falcon 40B Instruct, comparing their performance with both traditional statistical methods and other LLMs. To keep the figures comprehensible, we added only the following LLMs: Claude 3 Opus, Chat GPT, Mixtral 8x7B, and Mistral 7B.

We remark that the Falcon LLM team has released their training data,[21] offering further insights into how the training environments of contemporary LLMs can result into LLMs being capable of regression.

Due to the context size limitations of Falcon,[22] we only evaluated it on the linear regression datasets and on the `Original #1` dataset. The other datasets have a larger number of input variables (e.g., `Friedman #2` has 5 input variables) and we could not fit 50 in-context examples. We show our results on four datasets, `Regression NI 1/1`, `Regression NI 1/2`, `Regression NI 2/2`, `Original #1` in Figures 37, 38, 39 and 40. Additionally, we include the corresponding rank heatmap in Figure 41.

We make the following observations. First, Falcon 40B outperforms our unsupervised baselines. Second, Falcon 40B outperforms Gradient Boosting and Random Forests on `Regression NI 1/1`.

Overall, Falcon 40B displays, as well, the capability of doing regression when given in-context examples, albeit to a smaller degree than when compared to more powerful (and newer) models.

---

[21] https://huggingface.co/datasets/tiiuae/falcon-refinedweb
[22] The context size of Falcon is 2048.

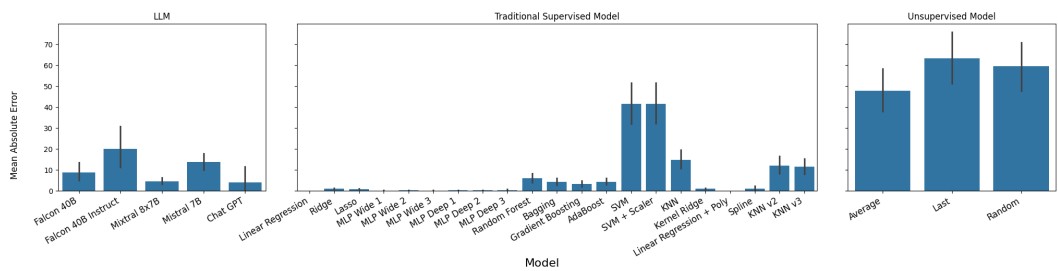

Figure 38: Falcon performance on the Regression NI 1/2 dataset

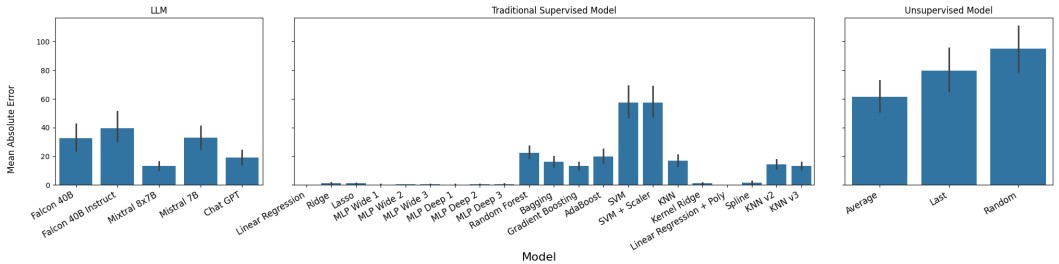

Figure 39: Falcon performance on the Regression NI 2/2 dataset

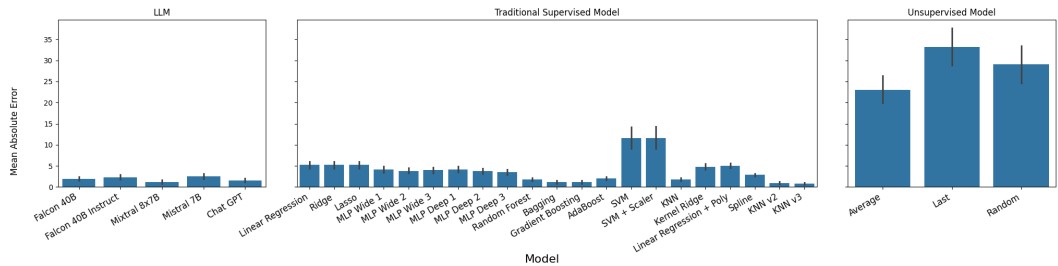

Figure 40: Falcon performance on the Original #1 dataset

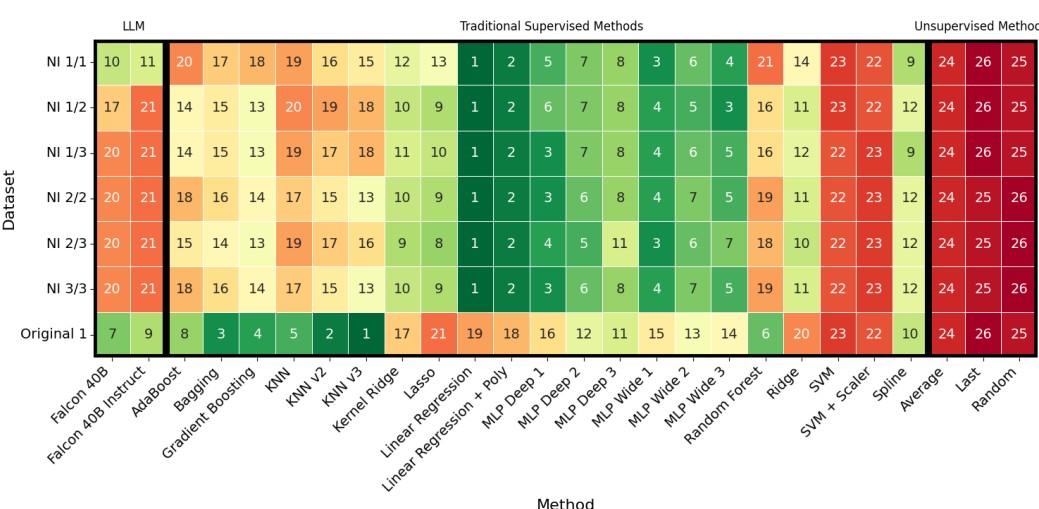

Figure 41: The rank of Falcon models, compared with traditional supervised methods and unsupervised heuristics. (best viewed in color)

### P.2 Performance When Knowing The Dataset Name

To further investigate potential data contamination, we conducted an experiment inspired by the methodology described in Golchin & Surdeanu (2024). This involves assessing model performance under two conditions: with and without explicit knowledge of the dataset being evaluated. Specifically, we modify the prompt shown in Figure 9 to mention the name of the dataset (e.g., `Friedman #1`, `Friedman #2`, `Friedman #3`), as detailed in Figure 42. This approach allows us to discern the impact of dataset awareness on the model's predictive accuracy, providing insights into the extent of potential contamination in the training data.

We present the comparative results in Table 14, which shows the average absolute error under conditions of dataset awareness versus unawareness. Notably, the mean absolute errors ($\downarrow$) remain closely matched across all scenarios. To statistically substantiate these observations, we performed paired t-tests for each dataset comparison. Given the multiplicity of tests performed, it became imperative to apply an adjustment for multiple comparisons to our p-values (Dunn, 1961; Benjamini & Hochberg, 1995). Following this adjustment, none of the p-values remained below the (typically used) 0.05 threshold, suggesting that the knowledge of the dataset name does not significantly affect model performance. Prior to adjustment, in two cases the resulting p-value was under 0.05: GPT-4 on `Friedman #3` (p-value 0.045) and Claude 3 Sonnet on `Friedman #3` (p-value 0.018). Note, however, that only in the case of GPT-4 was the performance corresponding to the Dataset Aware setting better. This analysis indicates that, within the bounds of statistical significance, there is no substantial evidence to suggest that the performance of the models is influenced by explicit knowledge of the dataset name, something which has been linked to contamination (Golchin & Surdeanu, 2024).[23]

> The task is to provide your best estimate for "Output" (Friedman #1). Please provide that and only that, without any additional text.
>
>
> Feature 0: . . .

Figure 42: The prompt we use to further investigate whether the large language models have seen instances of the datasets we tested them on.

---

[23]We investigated why the performance of Claude 3 Sonnet degraded on `Friedman #2` when given the dataset name. We found that it is (mostly) because of an outlier: the absolute difference between the model's prediction and expected output is $> 300$.

|  | Friedman #1 | Friedman #2 | Friedman #3 |
|---|---|---|---|
| MAE Dataset Aware | 2.820±2.332 | 39.086±64.787 | 0.089±0.143 |
| MAE Dataset Unaware | 2.798±2.242 | 32.787±66.165 | 0.102±0.156 |

(a) GPT-4

|  | Friedman #1 | Friedman #2 | Friedman #3 |
|---|---|---|---|
| MAE Dataset Aware | 1.933±1.401 | 5.435±7.460 | 0.074±0.163 |
| MAE Dataset Unaware | 2.002±1.470 | 6.374±10.440 | 0.070±0.148 |

(b) Claude 3 Opus

|  | Friedman #1 | Friedman #2 | Friedman #3 |
|---|---|---|---|
| MAE Dataset Aware | 2.118±1.791 | 11.332±33.119 | 0.095±0.175 |
| MAE Dataset Unaware | 2.192±1.748 | 5.375±5.940 | 0.088±0.166 |

(c) Claude 3 Sonnet

Table 14: Comparison of Mean Absolute Error (MAE ↓) for Models With and Without Dataset Name Awareness.

## Q  Does The Performance Plateau?

In the following, we investigate whether the performance of LLMs continues to improve as the number of in-context exemplars increases. Because this experiment is dependent on the context size of the LLM, this imposes additional constraints on which LLMs we can use. To this end, we experimented with the following dataset sizes: {20, 50, 60, 70, 80, 90, 100, 150, 200, 250, 300, 400, 500}.

We experiment with ChatGPT and GPT-4. For GPT-4, we found that the performance keeps improving with the number of in-context exemplars at least until 500. We present our results in Figures 43, 44, 45, 46, 47. We repeated each experiment 20 times for each dataset size. We report the mean and 95% confidence.

Aggregating the results, we observed that GPT-4 performed better than Random Forest in 92% of the cases, better than Gradient Boosting in 51% of the cases, and better than Linear Regression with Polynomial Features (`Linear Regression + Poly`) in 40% of the cases, across all 5 datasets and all dataset sizes.

For example, from Figure 44 we can observe that while `Linear Regression + Poly` performs much better than `GPT-4` in small data regimes, this performance gap decreases as the number of examples increases, suggesting that the model is indeed capable of leveraging a larger number of examples.

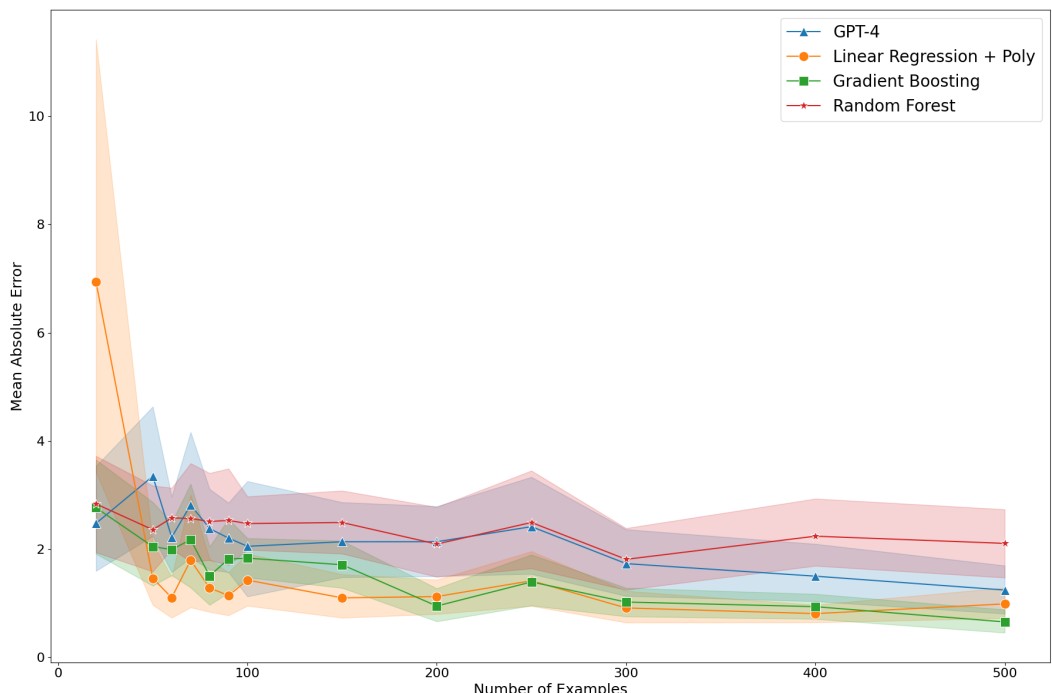

Figure 43: Performance comparison between GPT-4 and three traditional supervised models on the Friedman #1 dataset. The performance of GPT-4 remains good, outperforming Random Forest. (best viewed in color)

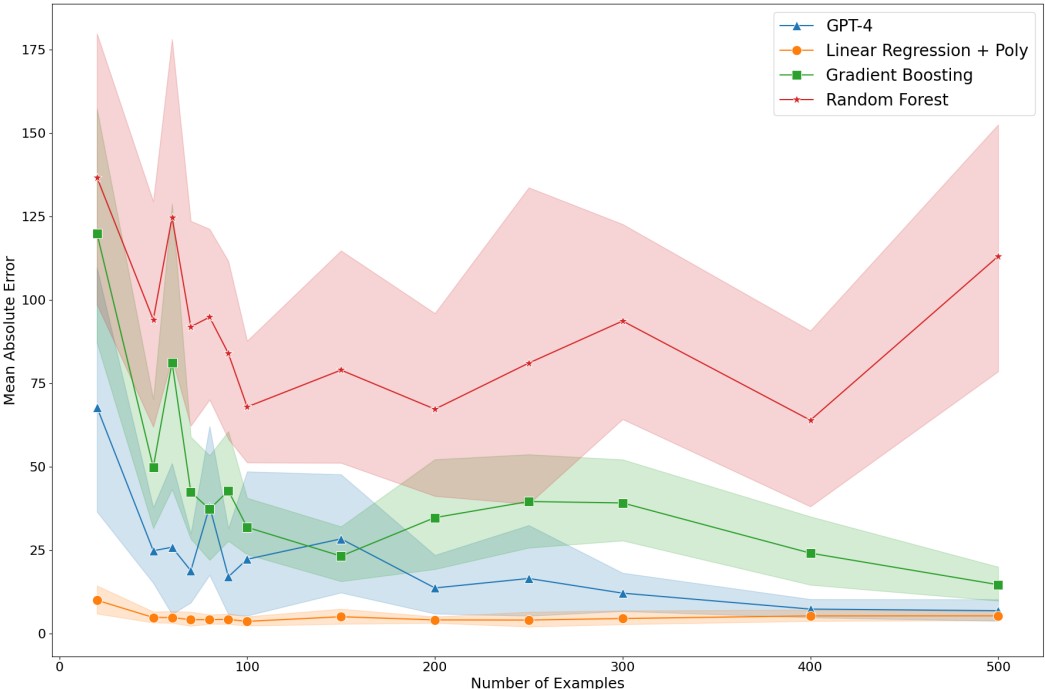

Figure 44: Performance comparison between GPT-4 and three traditional supervised models on the Friedman #2 dataset. The performance of GPT-4 improves with the number of examples given, approaching that of Linear Regression + Poly. (best viewed in color)

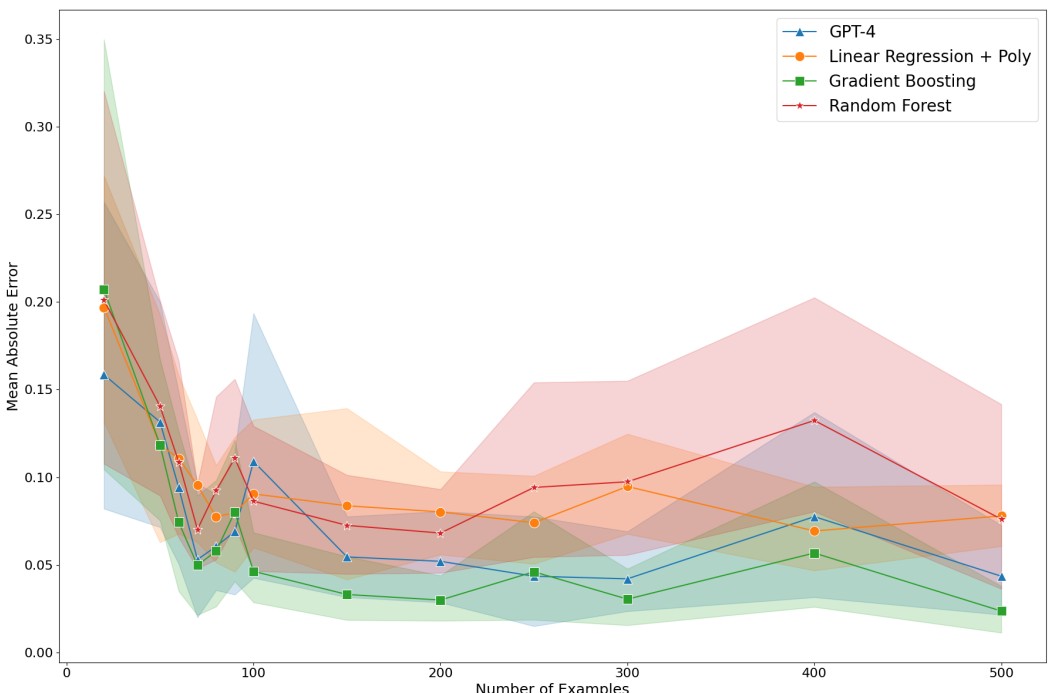

Figure 45: Performance comparison between GPT-4 and three traditional supervised models on the Friedman #3 dataset. (best viewed in color)

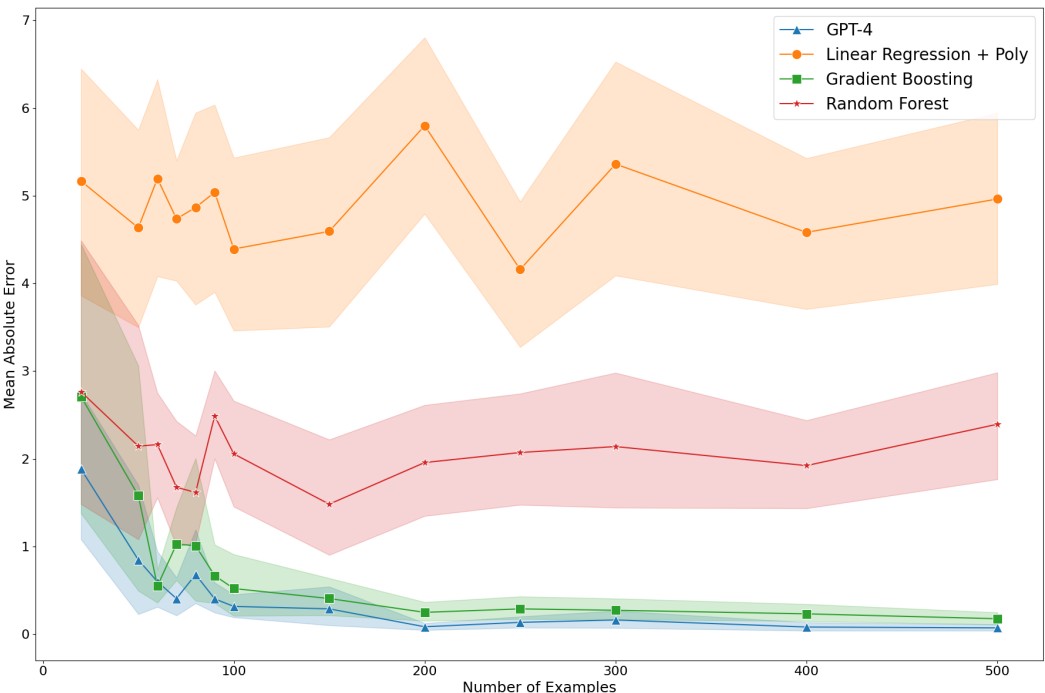

Figure 46: Performance comparison between GPT-4 and three traditional supervised models on the Original #1 dataset. The performance of GPT-4 increases with the number of examples given. (best viewed in color)

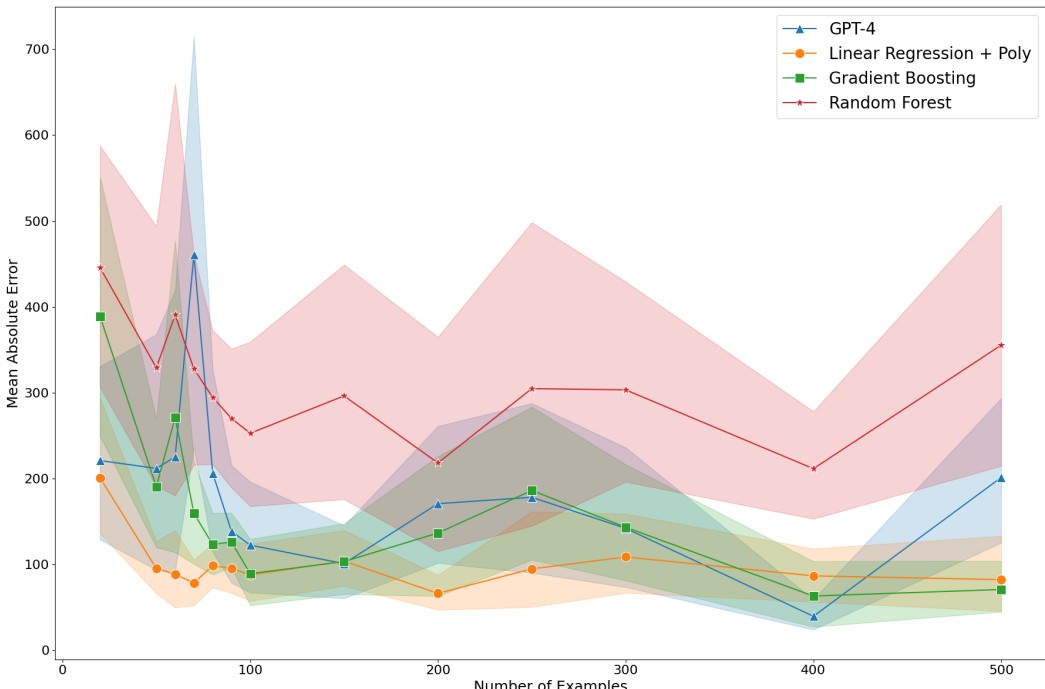

Figure 47: Performance comparison between GPT-4 and three traditional supervised models on the Original #2 dataset. The performance of GPT-4 is no worse to that of Random Forest (or, to a lesser extent, to that of Gradient Boosting) as the number of examples increases. (best viewed in color)

# R Beyond Transformer-Based LLMs

In our study, we initially focused on transformer-based large language models (LLMs). To broaden our scope, we explore the capabilities of non-transformer LLMs, including a RWKV-based 14B LLM (Peng et al., 2023) and with StripedHyena (Poli et al., 2023a;b), a 7B LLM. The performance rankings for these models, along with the transformer-based Mistral 7B for comparison, are illustrated in the heatmap provided in Figure 48. We tried running Falcon 7B as well, but it produced invalid outputs for almost all examples and all datasets, therefore we skip it. StripedHyena also encountered difficulties, producing invalid outputs in certain scenarios, such as 98% invalid responses for the Friedman #2 dataset. Consequently, we omitted Friedman #2 and Original #2 from its evaluation. However, it is important to highlight that the other models evaluated did not exhibit these issues and were able to generate valid outputs consistently. We make the following observations.

First, we remark that performance-wise, RWKV is worse than traditional transformer-based LLMs, although it generally remains better than our unsupervised baselines, with the exception on two linear regression datasets: Regression NI 1/3 and Regression NI 2/3. Nevertheless, we remark that on Original #1, RWKV outperforms many MLP variants, despite no gradient updates.

Second, we remark that the performance of Striped Hyena 7B is generally lower than than some of our unsupervised baselines. We note that there is a notable exception for Original #1. For this dataset, KNN approaches work well, as evident by the good performance obtained by nearest neighbor approaches.

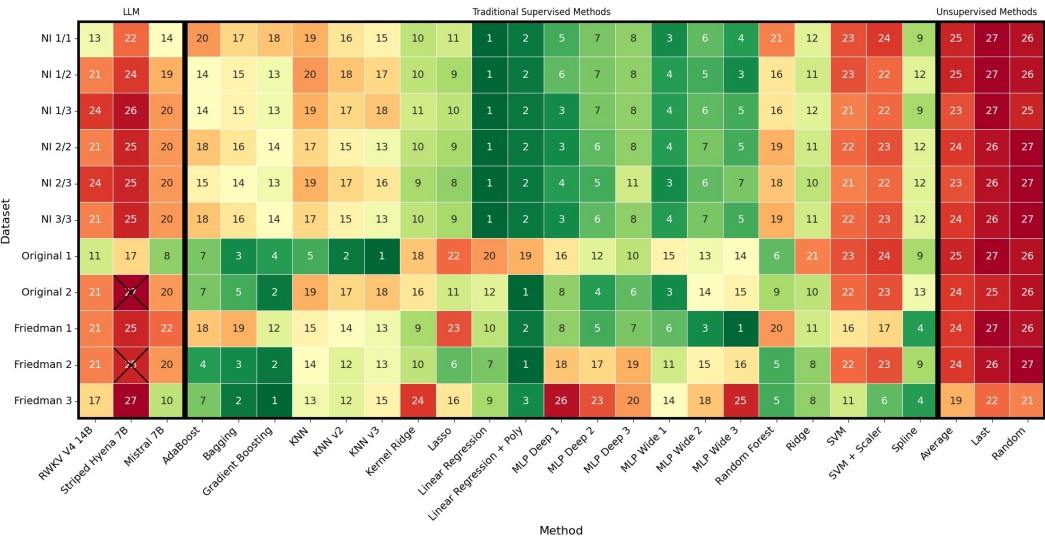

Figure 48: The ranks of RWKV V4 14B and (RWKV Architecture) and StripedHyena Nous 7B (Hyena Architecture) compared with traditional supervised method. We included Mistral 7B for comparison, as it is a model similar in size. Notably, RWKV's performance, while generally lower than that of transformer-based LLMs, still surpasses our unsupervised baselines. (best viewed in color)

