# OpenReview forum: "From Words to Numbers: Your Large Language Model Is Secretly A Capable Regressor When Given In-Context Examples"
_colmweb.org/COLM/2024/Conference — COLM_

### Official Review · Reviewer_n2HA · 2024-05-10

**Rating:** 5
**Confidence:** 3
**Ethics Flag:** 1

**Summary:**

This paper empirically compares the performance of LLMs on linear and non-linear regression tasks against traditional supervised models. The evaluations of LLMs are conducted in an ICL manner, where the training input and output are used as exemplars. Several different supervised methods (with their variations) are studied as the baselines, including linear regression, multi-layer perceptron, Random Forest, Bagging, Gradient Boosting, AdaBoost, SVM, KNN, Kernel Ridge, and Splines. To mitigate the risk of evaluation data being exposed in LLM pretraining, new non-linear regression datasets are also proposed in the evaluation. Overall, this study reveals that LLMs are capable of performing both linear and non-linear regression on the proposed datasets. Strong LLMs can outperform some supervised algorithms like linear regression and gradient boosting. They also approach the performance of the best supervised algorithms in some specific tasks.

**Questions To Authors:**

1. Bar plot might not be the best choice when the scale differences of MAEs are big (e.g., Figure 2a).
2. Sec. 2.1.2 how is $\epsilon$ determined in the experiments?
3. Sec. 2.1.2 re:
> When $d > 5$, the extra dimensions are ignored

why not just set $d=5$?

**Reasons To Accept:**

1. This paper studies an interesting task. While it is widely believed that LLMs are not good at math tasks, this paper revealed that strong LLMs can outperform simple supervised algorithms for certain problems. There is no widely accepted explanation for this yet, but it is an intriguing problem. From this perspective, the empirical findings from this paper are valuable.
2. I like the regret analyses in Sec. 5. It gave a rough estimation of how fast LLMs can learn to estimate the underlying functions. This can help determine the number of examples needed in different tasks.

**Reasons To Reject:**

Please correct me if my understanding is wrong. I think an important property missing in the paper is how the LLM’s performance change with different number of variables in the regression task. Based on Figure 2, there are at most 2 (informative) variables in the evaluations. This is significantly simpler than the dataset description of $1 \leq d \leq 20$ . Also, it is also not clear why there is no noise term $\epsilon * N(0, 1)$ in the linear regression task. With only 2 variables, it would be hard to claim that LLMs are capable of doing general linear regression tasks.

---

> ### Author Rebuttal · Authors · 2024-05-30
>
> Thank you for your review! We address each reason to reject and question below.
>
> Regarding the reason to reject:
>
> (1) We show in Figure 14 results with six variants, covering the possible combinations with 3 total variables. Our main motivations were to (i) keep the total number of tokens lower to accommodate a wider range of LLMs and (ii) reduce costs. To further supplement the results for linear regression, we conducted new experiments with: (i) 1 informative variable out of 10, (ii) 3 informative variables out of 10, and (iii) 5 informative variables out of 10. The performance is similar to NI 2/3 from Figure 14, where LLMs still perform reasonably well, but are outperformed by various baselines. For example, in NI 5/10, Claude 3 Opus outperforms methods like Gradient Boosting or Random Forest but lags behind methods like LR, which is expected given the nature of the datasets. For NI 3/10, Claude 3 Opus is outperformed by Random Forest and Gradient Boosting. Overall, these results align with those from Figure 14.
>
> We will update this in text.
>
>
> Regarding the Questions:
>
> (1) We included the rank heatmaps to address this issue. Additionally, we will include a table with average ranks across different dataset types. We also released our code and intermediate results, such that if any interested reader wants to compute other measures (or re-create ours), they will be able to do so. If the reviewer has other suggestions, we are willing to accommodate them.
>
> (2) In Sec 2.1.2 we included the definitions of each datasets in their most generic form (e.g., [make_friedman1](https://scikit-learn.org/stable/modules/generated/sklearn.datasets.make_friedman1.html)). In our experiments, we set $\epsilon$ to 0. We will add additional details in Appendix H to clarify this. Following the suggestions from the reviewers, we also included results with real-world datasets, which are expected to contain noise.
>
> (3) That detail is part of the Friedman #1 dataset definition (e.g., [make_friedman1](https://scikit-learn.org/stable/modules/generated/sklearn.datasets.make_friedman1.html)). We indeed used $d=5$, but included the generic definition of the dataset. We will further clarify this in text to avoid any confusion.

---

> > ### Comment · Reviewer_n2HA · 2024-06-05
> > **Response to the rebuttal**
> >
> > Thanks for the updates!
> >
> > I appreciate the extra effort in the additional experiments. I agree that the observations from this paper, although they are based-on low-dimension regression tasks, are valuable, and inspiring further research. But I do feel the several parts in the paper, (e.g., "Large Language Models Can Do Linear Regression", "Large Language Models Can Do Non-Linear Regression"), are a bit over-claiming.
> >
> > I raised my rating to 5.

---

> ### Author Response · Authors · 2024-06-05
> **Thank You for Your Feedback and Updated Rating; Additional Data and Insights Provided**
>
> Dear Reviewer,
>
> Thank you for your reply and for raising your score! We appreciate your engagement with our work and your constructive comments.
>
> We included a table with the average ranks across various types of datasets to summarize the results more concisely. We will include a similar table in the manuscript.
>
> More concretely, we show below a comparison between a subset of the models we used:
> - LLMs: 9 large language models (LLMs), both open and private:
>     - Open: DBRX, Mixtral 8x22b, Mixtral 8x7B
>     - Private: Claude 3 Opus, Claude 3 Sonnet, GPT-4, GPT-4 (20240409), Chat GPT, Gemini Pro
> - Traditional Supervised Methods: 5 traditional methods:
>     - Linear Regression + Poly, Linear Regression, Gradient Boosting, Random Forests
> - Unsupervised Methods: 3 unsupervised methods:
>     - Average, Random, Last
>
>
> We attached the table below (sorted by Overall).
>
> | Model Name | Average Rank Across Linear Datasets (6 datasets) | Average rank Across Original Datasets (5 datasets) | Average Rank Across Friedman Datasets (3 datasets) | Average Rank Across NN Datasets (2 datasets) | Average Rank Across Non-Linear Datastes (10 datasets) | Overall (16 datasets) |
> | -----------| ------------------------------------------------ | -------------------------------------------------- | -------------------------------------------------- | -------------------------------------------- | ----------------------------------------------------- | ------- |
> | Claude 3 Opus| 2.50  | 3.8| 2.00| 5.5| 3.6| 3.18|
> | Linear Regression + Poly| 2.33  | 6.4| 2.33| 2.5| 4.4| 3.62|
> | Claude 3 Sonnet| 5.33  | 4.0| 2.66| 7.0| 4.2| 4.62|
> | GPT-4| 5.00  | 5.8| 6.00| 8.0| 6.3| 5.81|
> | Linear Regression| 1.16  | 11.0| 9.00| 2.5| 8.7| 5.87|
> | GPT-4 (20240409)| 5.50  | 6.2| 6.00| 10.5| 7.0| 6.43|
> | Gradient Boosting| 9.50  | 5.6| 5.33| 2.0| 4.8| 6.56|
> | DBRX| 7.83  | 8.2| 8.66| 10.5| 8.8| 8.43|
> | Mixtral 8x22B| 9.66  | 7.0| 9.00| 9.0| 8.0| 8.62|
> | Gemini Pro| 7.66  | 7.6| 10.66| 12.0| 9.4| 8.75|
> | Random Forest| 12.33 | 8.8| 7.66| 5.5| 7.8| 9.50|
> | KNN| 12.66 | 10.2| 11.33| 3.0| 9.1| 10.43|
> | Mixtral 8x7B| 11.50 | 10.2| 12.33| 13.0| 11.4| 11.43|
> | Chat GPT| 12.00 | 13.0| 12.00| 15.0| 13.1| 12.68|
> | Average| 15.00 | 12.2| 15.00| 14.0| 13.4| 14.00|
> | Random| 16.50 | 16.6| 16.33| 16.5| 16.5| 16.50|
> | Last| 16.50 | 16.4| 16.66| 16.5| 16.5| 16.50|
>
>
> For example, Claude 3 Opus achieves an average rank of 2.5 on linear datasets, better than Random Forest, Gradient Boosting, or KNN, and only behind Linear Regression and Linear Regression + Poly.
> Similarly, it achieves an average rank of 3.6 on non-linear datasets, outperforming all the other methods in this comparison. Moreover, Claude 3 Opus achieves the best overall rank, outperforming all the other methods. Overall, at least in the settings we investigated, the LLMs indeed prove to be capable of regression.
>
>
> We acknowledge that there are open questions regarding the limits of LLMs in regression tasks, such as whether their strong performance is restricted to specific conditions like a small number of examples or dimensions. However, these questions should not detract from the merits of our findings. Instead, they highlight potential avenues for future research that our work enables.
>
> We believe that future LLMs could perform even better on regression tasks for the following reasons:
>
> - (i) Synthetic regression data might be included during pre-training to enhance the in-context learning algorithm.
> - (ii) Synthetic regression tasks could help evaluate how well LLMs leverage their contexts, with the expectation that error rates should predictably decrease with more examples if the LLMs can effectively use their context (therefore, performing well might become a target)

---

> > ### Author Response · Authors · 2024-06-06
> > **Regarding the over-claiming**
> >
> > We will provide our thoughts on this below.
> >
> > We understand your concern regarding the perceived over-claiming in some parts of the paper. Our intention was to highlight the surprising capability of LLMs to perform regression tasks, even if they do not always outperform traditional methods. The fact that LLMs can consistently outperform unsupervised baselines (Random, Average, Last) demonstrates their ability to handle regression tasks. Furthermore, we compared between LLMs and 70 variants of KNNs (Appendix K, expanded in revision). These KNN Regressor variants were created by modifying: (i) number of neighbors, (ii) weights (uniform or weighted), (iii) p.
> > We added our conclusion in the bullet point below. We argue that this is further evidence that the LLMs can, indeed, perform regression.
> >
> > - We observed that except on Friedman #1, for every other dataset, the top 9 best performing models are all LLMs. In other words, both closed-source models (e.g., Claude 3 Opus, GPT-4) and open-weights models (e.g., DBRX, Mixtral) outperform all KNN models on all the datasets except Friedman #1. This suggests that the mechanism implemented by in-context learning might be something more complex than KNN.
> >
> > Lastly, a low number of dimensions (i.e., $<10$) is reasonably common in practice. For example, from the UCI repository, there are 136 such datasets out of a total of 478 ([link](https://archive.ics.uci.edu/datasets?NumFeatures=0-10&skip=0&take=10&sort=asc&orderBy=NumInstances&search=&Types=Multivariate))
> >
> >
> > To summarize, the reasons we do not think this is overclaiming are: (1) The above experiment, comparing LLMs against 70 variants of KNNs, where the LLMs consistently outperform **all** KNN variants, (2) In almost all situations, the LLMs outperform our unsupervised baselines, (3) In many situations the LLMs outperform traditional supervised methods (Claude 3 Opus, for example, ranks 1st overall). While there might be certain settings where the LLMs will consistently under-perform even the unsupervised baselines (although we did not find any such examples), we believe that their strong performance, at least within the limits of our experimental setting, justifies the claim that they are indeed capable of regression. We believe this is a surprising and important observation.
> >
> > Nevertheless, we are open to revising the section names if there is consensus among the reviewers/AC that they may be perceived as over-claiming.

---

### Official Review · Reviewer_KJ4G · 2024-05-12

**Rating:** 6
**Confidence:** 4
**Ethics Flag:** 1

**Summary:**

This work conducts a detailed empirical analysis of the performance of several state-of-the-art pretrained Large Language Models for in-context learning of linear and non-linear regression. Around 11 of existing LLMs with 50 in-context exemplars are evaluated on synthetic datasets over 7 tasks comprising of Friedman regression benchmark, and linear and non-linear regression, along with regression with non-numerical inputs. The LLMs are compared with baseline methods of supervised methods from sklearn package such as linear regression (with lasso and ridge regularization), Bagging, GBDTs, Adaboost, etc, and unsupervised methods such as average and random. The synthetic data has inputs with dimensionality between 1-20, and around 50 in-context exemplars are used. The performance of the models is measured using Mean Average Error (MAE) and rank of the models. The authors expound on a sub-linear regret as the number of in-context exemplars scale.

The performance of these models is comparable to Random Forests and GBDTs, which are surprisingly poorer than linear and non-linear regression approaches. This could be the case as the dimensionality of the models is quite low (< 20). Since the datasets are synthetically generated from a generator function, the amount of data that can be generated an fed to train the supervised baselines are quite large in numbers. It would be useful to know how to extend these models to higher dimensionality of features, which is typically the case in real world scenarios, and  where Random Forest and GBDT baselines usually outperform other regression approaches. Given that baselines such as Garg et. al. 2022 are not considered for comparison, the nature of selecting the 50 in-context exemplars could play a vital role. Given the synthetic nature of the datasets, it should be possible to measure the variance of choosing a different subset of in-context exemplars.

**Reasons To Accept:**

1. The paper is well written, and well presented.

2. In-context learning is important to understand for capabilities of learning functions beyond natural language tasks, which this paper looks at.

3. The empirical analysis is extensive and covers a large number of LLMs and baseline methods.

**Reasons To Reject:**

1. The novelty is limited, the impact of which 50 in-context exemplars to select is not looked into. Domain adaptation of LLMs for in-context learning has already been studied in several works (Garg et al 2022, etc), and this work essentially works without any domain adaptation, and might not generalize to higher dimensions or other more complex datasets.

2. It is unclear how to extend this to work with higher dimensional datasets (>20) which are common for regression tasks.

3. More details on how long and many examples are used to train the baselines is important given the datasets are synthetic.

---

> ### Author Rebuttal · Authors · 2024-05-30
>
> Thank you for your review! We address each reason to reject below.
>
> (1) We argue that the novelty lies in demonstrating that off-the-shelf LLMs can perform regression reasonably well using in-context examples, without additional training.
> While the task of regression with transformers and in-context exemplars has been studied in previous work, they all trained the model for the specific task at hand. Our work shows that capable LLMs emerge with an underlying in-context learning algorithm capable of learning regression on-the-fly. This novel observation has interesting implications for future research.
>
> Regarding the impact of which 50 in-context exemplars are selected, the results in our paper were constructed by repeating each experiment 100 times, selecting train/test exemplars each time, and reporting the average.
>
> We did not claim that LLMs should be the go-to choice for any regression task. We acknowledge that LLMs currently have limitations. However, we demonstrate that off-the-shelf LLMs are more capable than previously thought. Their in-context learning algorithm, which emerges through pre-training, can perform regression tasks on-the-fly using in-context examples without any additional gradient updates.
> We believe our findings can contribute to a future theoretical framework by further describing transformers' capabilities. Additional implications include: (i) adding synthetic regression data during pre-training to improve the in-context learning algorithm; and (ii) testing how well LLMs use their context by evaluating their performance on regression tasks, where the error should predictably decrease with more examples if the LLM is indeed capable of using all its context well.
>
>
> (2) Regarding higher-dimensional datasets, we believe that techniques typically used for feature selection could be effectively applied to LLMs as well. We want to emphasize that off-the-shelf LLMs performing well even on lower-dimensional datasets is a significant finding in itself. This can pave the way for further exploration on higher-dimensional data.
>
> (3) We used the same examples for LLMs as for baselines, with baseline training handled by sklearn using default, battle-tested values. Similarly, no tuning was performed for LLMs to ensure a fair comparison. We will add more details in the Appendix B, where we describe the models we used.

---

> > ### Author Response · Authors · 2024-06-05
> > **Expanding on the Rebuttal**
> >
> > Dear Reviewer,
> >
> > We will provide an expanded discussion on the rebuttal.
> >
> > (1) Regarding the novelty, we argue that the novelty lies in the fact that, to the best of our knowledge, it has been unknown that off-the-shelf LLMs can do regression reasonably well, just by giving them in-context examples, without any additional training. While the task of regression with transformers and in-context exemplars has been studied in previous work, they all trained the model for the specific task at hand. Here, we show that capable LLMs already emerge with an underlying in-context learning algorithm that is capable of learning the regression task on-the-fly.
> >
> > We believe that the observation the off-the-shelf LLMs perform well even on synthetic datasets is a significant finding in itself and can pave the way for future work to: (i) better understand the limits of in-context learning, and (ii) develop better LLM-based approaches for regression. Other potential avenues for future work include: (i) adding synthetic regression data during pre-training to improve the in-context learning algorithm; and (ii) testing how well LLMs use their context by evaluating their performance on synthetic regression tasks, where the error should predictably decrease with more examples if the LLM is indeed capable of using all its context well.
> >
> > As you have acknowledged in your review, while domain adaption of LLMs has been studied, this work essentially works without any domain adaptation. We believe this is a novel observation.
> > > Domain adaptation of LLMs for in-context learning has already been studied in several works (Garg et al 2022, etc), and this work essentially works without any domain adaptation
> >
> > Regarding the impact of which 50 in-context exemplars are selected, we repeat each experiment 100 times and report the average. Each time, we select train/test exemplars. Therefore, we argue that our results are robust and representative.
> >
> > We discuss higher-dimensional datasets in (2). Regarding their applicability to more complex datasets, we would like to remark that our analysis encompassed the Friedman benchmark datasets. We leave the exploration of other datasets to future work.
> >
> > (2) Regarding higher-dimensional datasets, we believe that techniques typically used for feature selection could be effectively applied to LLMs as well. Although we agree that this remains to be proven, this can be a potential area for future research. Given the current manuscript's length of approximately 50 pages with the additional results, we leave this for future work.
> >
> > Our primary goal was not to propose LLMs as replacements for traditional regressors. Instead, we showed that LLMs are capable of performing regression, despite not being explicitly trained to do so. We want to emphasize that off-the-shelf LLMs performing well even on lower-dimensional datasets is a significant finding in itself.
> >
> > We did not claim that LLMs would necessarily perform well on higher-dimensional datasets. Our focus was on showing their potential in lower-dimensional scenarios, which we believe is significant in itself. Furthermore, while high-dimensional datasets are indeed common, there are many practical datasets with a modest number of features where our findings remain highly relevant.
> >
> > For example, on UCI datasets, there are 478 regression datasets ([link](https://archive.ics.uci.edu/datasets?skip=0&take=10&sort=desc&orderBy=NumHits&search=&Types=Multivariate)). Out of those, there are 136 with less than 10 features ([link](https://archive.ics.uci.edu/datasets?NumFeatures=0-10&skip=0&take=10&sort=desc&orderBy=NumHits&search=&Types=Multivariate)) and 206 with a number of features between 10 and 100 ([link](https://archive.ics.uci.edu/datasets?NumFeatures=10-100&skip=0&take=10&sort=desc&orderBy=NumHits&search=&Types=Multivariate)). Counting the ones with less than 20 features (inclusive), there are over 225 such datasets, just under 50% of the total number of datasets.
> > Therefore, even if LLMs might not generalize well to high-dimensional data (although this is not certain), there are still a large number of datasets that fall well within their realm of applicability.
> >
> > Lastly, we would like to emphasize that the contributions of this work should not be readily dismissed by speculation on how it would work in settings not investigated or mentioned in the paper.
> >
> > (3) We used exactly the same examples as for LLMs. The training of the baselines has been handled by sklearn with its default values, which are battle-tested and should provide a good start for most practitioners. Similarly, we did not perform any tuning for the LLMs either. All methods were used without any tuning to ensure a fair comparison. We will add more details in the Appendix B, where we describe the models we used. We also investigated in "Is It Just Better KNN?" (Appendix K) with a large number of different KNN variants. For example, Claude 3 Opus always outperform all 70 variants of KNNs we investigated.

---

> > > ### Comment · Reviewer_KJ4G · 2024-06-07
> > >
> > > I appreciate the additional real world evaluations and the detailed rebuttal from the authors. The empirical evaluations are detailed, the utility of applying this for real world tasks at high dimensional features can be further explored in current or future work. I raised the rating to 6 to reflect this.

---

> ### Author Response · Authors · 2024-06-05
> **Request for Feedback on Rebuttal and Consideration for Score Re-evaluation**
>
> Dear Reviewer,
>
> Since the discussion period will end soon, we would appreciate your feedback on our Rebuttal ([link](https://openreview.net/forum?id=LzpaUxcNFK&noteId=j9hEPsGrdM)) and the expanded points posted as a comment ([link](https://openreview.net/forum?id=LzpaUxcNFK&noteId=caL65oNRj8)). We believe we have thoroughly addressed all your points.
>
> If you find our responses satisfactory, we respectfully request you to reconsider your score.
>
> If there are any remaining issues or questions, we would be grateful for the opportunity to clarify them further.
>
> Thank you for your time and consideration.

---

### Official Review · Reviewer_Gng7 · 2024-05-12

**Rating:** 5
**Confidence:** 4
**Ethics Flag:** 1

**Summary:**

The paper evaluates how LLMs like GPT-4 perform linear and non-linear regression tasks using in-context examples, without further training. The authors present evidence that these models can match or exceed traditional supervised methods on several synthetic datasets. They explore how the performance of LLMs scales with the number of in-context examples and demonstrate that these models can achieve sub-linear regret, suggesting effective learning over time.

**Reasons To Accept:**

- The paper leverages existing LLM capabilities in a novel application, showing that these models can perform regression tasks effectively, which broadens the understanding of their potential uses.
- Demonstrating that LLMs can perform regression effectively without retraining could significantly reduce computational costs and streamline analytics processes in various fields.

**Reasons To Reject:**

- While synthetic datasets provide control over experimental conditions, the results might not generalize to real-world data, which can be noisier and less predictable.
- The paper does not extensively address whether the in-context learning approach could lead to overfitting, especially with an increasing number of context examples.
- The paper lacks a theoretical framework explaining why LLMs perform well in these tasks, which could strengthen the overall contribution.

---

> ### Author Rebuttal · Authors · 2024-05-30
>
> Thank you for your review! We address each reason to reject below.
>
> (1) While this choice has been deliberated, multiple reviewers have mentioned the lack of evaluation on real-world datasets as a weakness. Since future readers might wonder about the performance of LLMs on real datasets as well, we will add a new section titled “Performance On Real-World Datasets”.
>
> We ran new experiments on:
> - (1) Liver Disorders (UCI id=60)
> - (2) Real Estate (UCI id=477)
> - (3) Diabetes (from sklearn)
> - (4) Servo (UCI id=87)
>
> Each experiment was repeated 50 times, each time with different train/test splits. Results:
>
> - On 1, GPT-4 ranks 6, outperforming Gradient Boosting (GB) and Random Forest (RF); Claude 3 Opus performs worse, ranking 23rd;
> - On 2, GPT-4 ranks 6 and Claude 3 Opus ranks 4th. GB ranks 1st and RF ranks 9th;
> - On 3, GPT-4 ranks 16th and Claude 3 Opus ranks 19th. RF ranks first;
> - On 4, Claude 3 Opus 2nd. Both GPT-4 and Claude 3 Opus outperform RF. For this dataset, we only tested the LLMs against GB and RF.
>
> Overall, the strong performance of LLMs persists over real-world datasets as well. Tables skipped due to length constraints.
>
> Lastly, we did not intend to suggest LLMs should replace traditional supervised methods for regression. We believe that the observation the off-the-shelf LLMs perform well even on synthetic datasets is a significant finding in itself.
>
> (2) We do not expect overfitting to be an issue with more examples. Overfitting in general and its effect on in-context learning vs traditional methods is interesting but outside the scope of this paper (now ~50 pages with additional results). In Appendix L (Figures 31-35), we show GPT-4's performance improves with more examples (20-500), similar to other methods.
>
> (3) We would like to clarify that this paper is not intended to explain why ICL works in general or for regression specifically. Instead, our focus is on exploring the extent to which LLMs can perform regression. We believe our findings can contribute to a future theoretical framework by further describing transformers' capabilities. We hypothesize that LLMs emerge from pre-training as highly capable online meta-learners, able to perform tasks beyond those seen during pre-training. This hypothesis motivated Section 5.

---

> > ### Author Response · Authors · 2024-06-05
> > **Expanding on the Rebuttal**
> >
> > Dear Reviewer,
> >
> > We will provide an expanded discussion on the rebuttal.
> >
> >
> > (1) Please refer to our comment ([link](https://openreview.net/forum?id=LzpaUxcNFK&noteId=EZ3Xl5zZan)). We included 5 additional real-world datasets.
> >
> > (2) We do not expect overfitting to be problematic in the regime of an increasing number of examples. Regarding overfitting in general and how it affects in-context learning vs traditional methods, we believe this is an interesting question, but it is outside the scope of the paper (which already grew to ~50 pages with the additional results and experiments). We explore in Appendix L (Figures 31,.., 35) how the performance of GPT-4 compares with Gradient Boosting, Random Forest and LR + Poly when given more examples (20 <= x <= 500). We observed that GPT-4 behaves “normally”, in the sense that its performance improves with the number of examples at least as well as the other methods.
> >
> > Additional evidence that the LLMs are capable of leveraging can be found in the literature [1, 2].
> >
> >
> > (3) We would like to remark that this paper has not intended to be an explanation on why ICL works in general or why it works for regression. We included in the revision an extended abstract in the Appendix, expanding on the papers included in the Related Work, some of which attempt a theoretical understanding of the ICL mechanism. We would like to highlight that even though in-context learning took off with GPT-3 and many papers attempted to explain it [3, 4, 5, inter alia], the exact mechanisms at play in real-world LLMs, such as the ones we used in this work, are still not fully understood [6].
> >
> > Furthermore, we want to emphasize that the primary focus of our paper is on exploring the extent to which transformers can perform regression, rather than providing a theoretical framework for in-context learning (ICL).
> > We believe that our findings can be helpful for a future theoretical framework, as it further describes what transformers can do. The fact that off-the-shelf LLMs are capable of performing regression has not been previously known.
> > This discovery is particularly surprising because it is not limited to proprietary models like GPT-4 and Claude 3; it also extends to open-weight models such as Yi 34B, DBRX (Appendix), and Mixtral 8x22B (added in the Revision). This suggests that, perhaps, once an LLM reaches a certain level of capability, it might universally acquire the ability to perform regression, despite not being explicitly trained to do so.
> >
> > In our paper, we hypothesize that LLMs emerge from their pre-training as highly capable online meta-learners. This capability may explain their ability to perform a wide variety of tasks, including some that were not encountered during their pre-training phase, such as our original regression datasets. This hypothesis motivated the exploration in Section 5.
> >
> >
> >
> >
> > [1] [Agarwal, Rishabh et al. “Many-Shot In-Context Learning.” ArXiv abs/2404.11018 (2024): n. pag.](https://arxiv.org/pdf/2404.11018)
> >
> > [2] [Jiang, Yixing et al. “Many-Shot In-Context Learning in Multimodal Foundation Models.” (2024).](https://arxiv.org/pdf/2405.09798)
> >
> > [3] [Oswald, Johannes von et al. “Transformers learn in-context by gradient descent.” International Conference on Machine Learning (2022).](https://arxiv.org/pdf/2212.07677)
> >
> > [4] [Xie, Sang Michael et al. “An Explanation of In-context Learning as Implicit Bayesian Inference.” ArXiv abs/2111.02080 (2021): n. pag.](https://arxiv.org/abs/2111.02080)
> >
> > [5] [Ahn, Kwangjun et al. “Transformers learn to implement preconditioned gradient descent for in-context learning.” ArXiv abs/2306.00297 (2023): n. pag.](https://arxiv.org/pdf/2306.00297)
> >
> > [6] [Shen, Lingfeng et al. “Do pretrained Transformers Learn In-Context by Gradient Descent?” (2023).](https://arxiv.org/pdf/2310.08540)

---

> > ### Comment · Reviewer_Gng7 · 2024-06-06
> >
> > Thanks for your response. While I am still skeptical about the motivation, I'd like to raise my score to reflect the efforts the authors have made during the rebuttal period.

---

> ### Author Response · Authors · 2024-06-05
> **Request for Feedback on Rebuttal and Consideration for Score Re-evaluation**
>
> Dear Reviewer,
>
> Since the discussion period will end soon, we would appreciate your feedback on our Rebuttal ([link](https://openreview.net/forum?id=LzpaUxcNFK&noteId=onU0uvgdVQ)) and the expanded points posted as a comment ([link](https://openreview.net/forum?id=LzpaUxcNFK&noteId=ZgNCzfxwhA)). We believe we have thoroughly addressed all your points.
>
> If you find our responses satisfactory, we respectfully request you to reconsider your score.
>
> If there are any remaining issues or questions, we would be grateful for the opportunity to clarify them further.
>
> Thank you for your time and consideration.

---

> ### Author Response · Authors · 2024-06-06
> **Reply to Reviewer Gng7**
>
> Thank you for raising your score and engaging with us!
>
> We would like to further elaborate on the motivation of this paper.
>
> Understanding the underlying mechanism of In-Context Learning (ICL) has spawned many recent works. Importantly, these works predominantly train a model specifically to do ICL. We complement these works by exploring the performance of off-the-shelf Large Language Models (LLMs) without any further training. This is important for the following reasons: (i) ICL emerges in LLMs without them being specifically trained for it, and (ii) it is more practical, as practitioners will typically use off-the-shelf LLMs rather than training one from scratch.
>
> Furthermore, as you noted in the "Reasons to Accept" section of the original review, our work broadens the understanding of their potential uses and could streamline analytics processes in various fields.
>
> In other words, the motivation of this work is to further enhance our understanding of the underlying mechanisms at play in ICL. We expand upon previous work aimed at improving our understanding of ICL by investigating it under a more realistic scenario, with off-the-shelf LLMs.

---

### Official Review · Reviewer_y3a3 · 2024-05-14

**Rating:** 8
**Confidence:** 4
**Ethics Flag:** 1

**Summary:**

The paper looks at the ability of 11 pre-trained LLMs like GPT-4, Claude 3, and others to perform both linear and non-linear regression tasks using in-context learning (ICL). The results show that some LLMs can perform on par or outperform traditional supervised methods such as Random Forest, Gradient Boosting, and AdaBoost on challenging synthetic regression datasets - without additional training or parameter updates. Additionally, the paper explores how the performance of LLMs scales with the number of in-context examples, empirically showing that some LLMs obtain sub-linear regret.

**Reasons To Accept:**

* This paper is clearly written, well motivated and easy to follow.
* It nicely expands on earlier work for Transformers and ICL with off-the-shelf LLMs.
* It conducts a thorough empirical analysis with clear experimental setup for reproducibility and strong baselines for comparison.
* The analysis of scaling performance and use of regret shows how many examples are needed to get a certain level of performance.
* The findings have practical use, suggesting that LLMs can potentially reduce the need for traditional supervised learning methods in certain regression scenarios.

**Reasons To Reject:**

* Because of the nature of the experiments on synthetic data to avoid encountering data that may have been in the training set, there are no experiments with benchmark/ real world regression problems.
* In the last study on regret, it is not clear how regret correlates to MAE, and why the regret values or so different between the different models evaluated.
* It does not consider the computational requirements for scaling these LLMs to even larger datasets or higher-dimensional regression tasks, leaving a question of practical feasibility for real-world applications of larger problems.

---

> ### Author Rebuttal · Authors · 2024-05-30
>
> Thank you for your review! We address each reason to reject below.
>
> (1) While this choice has been deliberated, multiple reviewers have mentioned the lack of evaluation on real-world datasets as a weakness. Since future readers might wonder about the performance of LLMs on real datasets as well, we will include a new section titled “Performance On Real-World Datasets”.
>
> We ran new experiments on the following datasets:
> - (1) Liver Disorders (UCI id=60)
> - (2) Real Estate (UCI id=477)
> - (3) Diabetes (from sklearn)
> - (4) Servo (UCI id=87)
>
> Each experiment was repeated 50 times, each time with different train/test splits. Results:
>
> - On 1, GPT-4 ranks 6, outperforming Gradient Boosting and Random Forests; Claude 3 Opus performs worse, ranking 23rd;
> - On 2, GPT-4 ranks 6 and Claude 3 Opus ranks 4th. Gradient Boosting ranks 1st and Random Forest ranks 9th;
> - On 3, GPT-4 ranks 16th and Claude 3 Opus ranks 19th. Random Forests ranks first;
> - On 4, this dataset contain features both categorical and numerical and we tested only Gradient Boosting and Random Forests. Claude 3 Opus 2nd. Both GPT-4 and Claude 3 Opus outperform Random Forests.
>
> Overall, the strong performance of LLMs persists over real-world datasets as well. Tables skipped due to length constraints.
>
> (2) In the regret experiment, we wanted to analyze how well the LLM are capable of adapting when increasing the number of in-context examples. Borrowing from online learning, ideally, a capable model would be able to achieve the performance of a model that has hindsight information, fast. Our motivation was our hypothesis that, perhaps, LLMs emerge from their pre-training as very capable online meta-learners, which is why they are capable of performing so many tasks, some of which were not seen during their pre-training phase (like our Original regression datasets). In Figure 7, for example, the Y-scale is different between the two figures because the datasets are different. Original #1, for example, has 0 <= f(x) <= 100. Friedman #2, on the other hand, can have values larger than 1000.
>
> (3) We agree and will include this in a dedicated section for limitations. We would like to remark that we did not intend to suggest that LLMs should be used, at least currently, for regression instead of more traditional methods like Gradient Boosting.

---

> ### Author Response · Authors · 2024-06-05
> **Thank you for your Feedback**
>
> Dear Reviewer,
>
> Thank you for your positive review and the high score. We are glad that you found our work valuable.
>
> If you have any further questions or need additional clarifications, please let us know. We would be happy to provide more information.
>
> Thank you again for your thoughtful feedback.

---

### Author Response · Authors · 2024-06-01
**Results on Real-World Datasets**

We present the new results obtained from testing the LLMs on real-world datasets. Two reviewers highlighted the absence of real-world evaluations, and we recognize that future readers may also wonder about the LLMs' performance on real-world datasets. To address this, we ran additional experiments on the following datasets.

- (1) Liver Disorders (UCI id=60)

| Dataset         | Method                   | MAE  | Rank |
| --------------- | ------------------------ | ---- | ---- |
| Liver Disorders | GPT-4                    | 2.55 | 6.0  |
| Liver Disorders | Gradient Boosting        | 2.56 | 8.0  |
| Liver Disorders | Random Forest            | 2.61 | 12.0 |
| Liver Disorders | Linear Regression + Poly | 2.67 | 19.0 |
| Liver Disorders | Average                  | 2.79 | 22.0 |
| Liver Disorders | Claude 3                 | 3.04 | 24.0 |
| Liver Disorders | Random                   | 3.23 | 27.0 |
| Liver Disorders | Last                     | 3.46 | 28.0 |



- (2) Real Estate (UCI id=477)

| Dataset     | Method                   | MAE   | Rank |
| ----------- | ------------------------ | ----- | ---- |
| Real Estate | Gradient Boosting        | 4.30  | 1.0  |
| Real Estate | Claude 3                 | 5.08  | 4.0  |
| Real Estate | GPT-4                    | 5.254 | 6.0  |
| Real Estate | Linear Regression + Poly | 5.37  | 7.0  |
| Real Estate | Random Forest            | 5.72  | 10.0 |
| Real Estate | Average                  | 11.12 | 26.0 |
| Real Estate | Random                   | 14.38 | 27.0 |
| Real Estate | Last                     | 15.26 | 29.0 |

- (3) Diabetes (from sklearn)

| Dataset  | Method                   | MAE    | Rank |
| -------- | ------------------------ | ------ | ---- |
| Diabetes | Random Forest            | 41.30  | 1.0  |
| Diabetes | Gradient Boosting        | 47.95  | 16.0 |
| Diabetes | GPT-4                    | 50.26  | 17.0 |
| Diabetes | Claude 3                 | 54.13  | 19.0 |
| Diabetes | Linear Regression + Poly | 55.84  | 20.0 |
| Diabetes | Average                  | 68.02  | 25.0 |
| Diabetes | Random                   | 94.34  | 27.0 |
| Diabetes | Last                     | 103.96 | 28.0 |

- (4) Servo (UCI id=87)

| Dataset | Method                   | MAE  | Rank |
| ------- | ------------------------ | ---- | ---- |
| Servo   | Gradient Boosting        | 0.24 | 2.0  |
| Servo   | Claude 3                 | 0.27 | 4.0  |
| Servo   | GPT-4                    | 0.43 | 10.0 |
| Servo   | Random Forest            | 0.44 | 11.0 |
| Servo   | Linear Regression + Poly | 0.62 | 13.0 |
| Servo   | Average                  | 1.42 | 25.0 |
| Servo   | Last                     | 1.60 | 27.0 |
| Servo   | Random                   | 1.77 | 29.0 |

- (5) Movies (UCI id=424)

| Dataset | Method                   | MAE (e+07) | Rank |
| ------- | ------------------------ | ---------- | ---- |
| Movies  | Random Forest            | 3.02       | 2.0  |
| Movies  | Gradient Boosting        | 3.02       | 3.0  |
| Movies  | Claude 3                 | 4.50       | 17.0 |
| Movies  | GPT-4                    | 5.47       | 20.0 |
| Movies  | Average                  | 6.19       | 21.0 |
| Movies  | Linear Regression + Poly | 6.39       | 22.0 |
| Movies  | Random                   | 8.16       | 23.0 |
| Movies  | Last                     | 8.25       | 24.0 |





Except (4) Servo and (5) Movies, all the other datasets have numeric features. (4) Servo has categorical features. (5) Breast Cancer has categorical features and missing features. The LLMs can implicitly handle both: (i) categorical features and (ii) missing features without any additional pre-processing. For missing features, we simply put "unknown".
For the traditional supervised methods, we encoded the categorical features with One Hot Encoder. For missing features, we imputed them with the mean. We ran all the traditional supervised methods described in the paper.

Lastly, we do not intend to suggest LLMs should replace traditional supervised methods for regression or that they obtain a superior performance in all situations. We believe that the observation the off-the-shelf LLMs perform well even on synthetic datasets is a significant finding in itself and can pave the way for future work to: (i) better understand the limits of in-context learning, and (ii) develop better LLM-based approaches for regression. Other potential avenues for future work include: (i) adding synthetic regression data during pre-training to improve the in-context learning algorithm; and (ii) testing how well LLMs use their context by evaluating their performance on synthetic regression tasks, where the error should predictably decrease with more examples if the LLM is indeed capable of using all its context well.

---

### Author Response · Authors · 2024-06-05
**Open Source Code and Intermediate Results; Summary of Changes**

We open sourced our code together with the intermediate results and examples of how to re-create some of the plots in the paper. This way, if anyone is interested in performing other analysis or using other metrics, they can do so.

We mentioned this in our rebuttal. In this comment, we included the Anonymous Github [link](https://anonymous.4open.science/r/llm4regression-BD81/README.md). The Github page also contains: (1) Local Jupyter notebooks, (2) Links to Google Colab notebooks, (3) Examples on how to read the data and re-construct some of the plots in the paper. We believe (1) and (2) will be especially helpful, allowing interested readers to quickly try do to regression with LLMs.

We also made small changes to the Appendix. Our motivation for including these changes is to further strengthen the paper. These changes complement the paper without altering its main conclusions or the overall message. The changes include:
- Small reorganization to ensure that relevant figures stay closer to the corresponding sections
- Expanding on the number of original synthetic datasets
- A full section in the Appendix titled "Could It Be Just Contamination?" where we expand on the measures we took to ensure that the results presented in the paper are not an artifact of contamination. This is has also been a motivation for including more "Original" synthetic datasets. Additionally, we also:
(1) included LLMs with known training data (Falcon), and
(2) employed techniques from the literature to detect contamination in black-box models.
- A full section in the Appendix titled "Beyond Transformer-Based LLMs", where we investigate the performance of LLMs with architectures different from transformers. Specifically, we used `RWKV-based 14B` and `StripedHyena 7B`. We found that RWKV is worse than traditional transformer-based LLMs, although it generally remains better than our unsupervised baselines, with the exception on two linear regression dataset.

---

### Author Response · Authors · 2024-06-07
**Thank you to all Reviewers**

Since the discussion period will end soon, we want to thank the reviewers again for their feedback and engagement.
We are glad that the reviewers appreciated the novelty of our work [Reviewer Gng7], its thorough empirical analysis [Reviewers y3a3, KJ4G], found it well motivated [Reviewer y3a3]. The regret analysis presented in Section 5 was praised as well [Reviewers y3a3, n2HA].
Additionally, it was judged to broaden the understanding of ICL [Reviewer Gng7]. Reviewers also found our work to be well-written [Reviewers y3a3, KJ4G], practical and useful [Reviewer y3a3, Gng7].


We will summarize below the steps we took to ensure that what we have presented in the main manuscript is robust.

Our findings suggest that LLMs are indeed capable of doing regression. Besides the main highlights shown in the main manuscript, we took many precautions to ensure that what we have observed is not simply an artifact of our design choices. The details of these experiments can be found in the Appendix. To this end:
- All the results we represent an average of as many as 100 runs; For example, all the results in Section 3 and 4 are with an average of 100 runs with different train-test splits, ensuring robustness;
- Experimented with 5 original datasets (post revision; 2 in the original submission) to show that what we have seen is not simply because the LLMs saw that data on the Internet
    - We also added a new section in the Appendix titled "Could It Be Just Contamination?" where we discuss the contamination problem even further by: (1) using LLMs with known training data (Falcon 40B) and (2) employing data contamination checks
-  Increased the number of in-context examples given to the LLM to 500 to ensure that this strong performance is not present only in low-data regimes (Appendix L, "Does The Performance Plateau?")
- Compared the performance against 70 different variants of KNNs to test whether the LLMs are just better KNNs or not (Appendix K "Is It Just Better KNN?", extended post revision).
- We ablated the choice of rounding the numbers to two digits (Appendix J, "Effects of Rounding")

In all the experiments we performed, the same conclusion prevailed: LLMs are indeed capable of performing regression, with some LLMs being more effective than others (e.g., Claude 3 Opus).



We believe we have adequately addressed your concerns and suggestions. We are happy to incorporate any additional feedback, especially with the availability of an extra page.


As detailed in the individual comments, we will make the following changes:
1. Section titled "Performance on Real-World Datasets", which will contain a discussion of the performance of LLMs on real world datasets. Following the reviewer's suggestions, we provided results on 5 real-world datasets in our rebuttal (please see the relevant comment [here](https://openreview.net/forum?id=LzpaUxcNFK&noteId=EZ3Xl5zZan))
2. A section dedicated to limitations of using LLMs for regression in general and of this work in particular
3. Improve the clarity of certain sections, as mentioned by the reviewers, and include the clarification experiments.

---

### Decision · Program_Chairs · 2024-07-10

**Decision:**

Accept

**Comment:**

This paper demonstrates several LLMs can be effective at performing regressions in-context, in some cases outperforming supervised methods.

The reviewers noted several strengths of the paper:
- The paper makes an interesting point about the emergent capabilities of LLMs.
- The paper provides an important expansion on prior work by studying pretrained LLMs rather than ones trained specifically to be able to do ICL.
- The paper is clearly written.
- The experimental setup is strong, with clear baselines, a comprehensive set of experiments, and enough information provided to make for strong reproducibility

Several limitations also came up:
- The paper does not provide a theoretical understanding of why LLMs can perform regression tasks.
- LLMs using ICL may still not be effective in real-world regression tasks or as replacements for supervised methods
- There are unexplored directions about several details of the experiments such as the number of in-context examples and the numbere of dimensions.

However, I found the following comment from the author response to be compelling in response to the first and third points:  “We acknowledge that there are open questions regarding the limits of LLMs in regression tasks, such as whether their strong performance is restricted to specific conditions like a small number of examples or dimensions. However, these questions should not detract from the merits of our findings. Instead, they highlight potential avenues for future research that our work enables.” And regarding the second point, I was convinced by the authors’ response that these directions are not the goal of the work (the goal is to understand LLMs, not to propose a new approach for doing regressions).

Overall, I am recommending acceptance due to the interesting results that make important points about ICL in LLMs, and due to the fact that I found the response from the authors to be generally compelling in addressing reviewer concerns. The response from the authors also added some new results with real-world datasets that will enhance the paper.

I thank the reviewers and authors for a very productive discussion period.